# Crystal structure of the α$_{1B}$-adrenergic receptor reveals molecular determinants of selective ligand recognition

Mattia Deluigi [1], Lena Morstein [1,10], Matthias Schuster[2,10], Christoph Klenk [1], Lisa Merklinger[1,7],
Riley R. Cridge[3], Lazarus A. de Zhang[3,4], Alexander Klipp [1,8], Santiago Vacca [1], Tasneem M. Vaid[5],
Peer R. E. Mittl [1], Pascal Egloff [1], Stefanie A. Eberle[1,9], Oliver Zerbe [2], David K. Chalmers [6],
Daniel J. Scott[3,4] ✉ & Andreas Plückthun [1] ✉

α-adrenergic receptors (αARs) are G protein-coupled receptors that regulate vital functions of the cardiovascular and nervous systems. The therapeutic potential of αARs, however, is largely unexploited and hampered by the scarcity of subtype-selective ligands. Moreover, several aminergic drugs either show off-target binding to αARs or fail to interact with the desired subtype. Here, we report the crystal structure of human α$_{1B}$AR bound to the inverse agonist (+)-cyclazosin, enabled by the fusion to a DARPin crystallization chaperone. The α$_{1B}$AR structure allows the identification of two unique secondary binding pockets. By structural comparison of α$_{1B}$AR with α$_2$ARs, and by constructing α$_{1B}$AR-α$_{2C}$AR chimeras, we identify residues 3.29 and 6.55 as key determinants of ligand selectivity. Our findings provide a basis for discovery of α$_{1B}$AR-selective ligands and may guide the optimization of aminergic drugs to prevent off-target binding to αARs, or to elicit a selective interaction with the desired subtype.

[1] Department of Biochemistry, University of Zurich, Winterthurerstrasse 190, CH-8057 Zurich, Switzerland. [2] Department of Chemistry, University of Zurich, Winterthurerstrasse 190, CH-8057 Zurich, Switzerland. [3] The Florey Institute of Neuroscience and Mental Health, The University of Melbourne, 30 Royal Parade, Parkville, VIC 3052, Australia. [4] Department of Biochemistry and Pharmacology, The University of Melbourne, Parkville, VIC 3010, Australia. [5] Department of Pharmaceutical Sciences, University of Illinois at Chicago, Chicago, IL, USA. [6] Monash Institute of Pharmaceutical Sciences, Monash University, 381 Royal Parade, Parkville, VIC 3052, Australia. [7] Present address: Department of Biotechnology and Biomedicine, Technical University of Denmark, Søltofts Plads, 2800 Kgs Lyngby, Denmark. [8] Present address: Department of Chemistry and Applied Biosciences, ETH Zurich, Vladimir-Prelog-Weg 1–5/10, CH-8093 Zurich, Switzerland. [9] Present address: Department of Biomedical Sciences, Faculty of Health and Medical Sciences, University of Copenhagen, Blegdamsvej 3B, 2200 Copenhagen, Denmark. [10] These authors contributed equally: Lena Morstein, Matthias Schuster. ✉ email: daniel.scott@florey.edu.au; plueckthun@bioc.uzh.ch

Adrenergic receptors (ARs), or adrenoceptors, are aminergic G protein-coupled receptors (GPCRs) subdivided into nine distinct subtypes in humans[1,2]—three $\alpha_1$ARs ($\alpha_{1A}$, $\alpha_{1B}$, $\alpha_{1D}$), three $\alpha_2$ARs ($\alpha_{2A}$, $\alpha_{2B}$, $\alpha_{2C}$), and three $\beta$ARs ($\beta_1$, $\beta_2$, $\beta_3$). ARs are widely yet differentially expressed in the central nervous system (CNS) and in peripheral sympathetic nerves, as well as in sympathetically innervated tissues throughout the body[3,4]. Upon activation by the endogenous catecholamines epinephrine and norepinephrine, ARs mediate a large variety of physiological functions, many of which are of considerable clinical relevance[4]. For instance, drugs blocking $\beta_1$AR are widely prescribed to treat hypertension and heart failure, while $\beta_2$AR agonists are used as bronchodilators in asthma therapy[4,5].

In contrast to $\beta$ARs, the therapeutic potential of $\alpha$ARs is largely unexploited for at least two reasons. First, there is a lack of truly selective ligands for individual $\alpha_1$AR and $\alpha_2$AR subtypes, which often mediate opposing physiological functions[6,7]. As a consequence, drugs that act through $\alpha$ARs are only second-line agents to treat hypertension, pain, or neuropsychiatric disorders, due to their off-target side effects and limited therapeutic benefits[6]. Second, there is an incomplete understanding of the individual $\alpha$AR subtypes' physiological and pathophysiological roles, as the lack of selective compounds has hampered research. Nonetheless, studies in transgenic mice indicate that distinct $\alpha_1$AR subtypes mediate different functions in the heart, CNS, and urogenital system[7–9]. Stimulation of $\alpha_{1A}$AR and $\alpha_{1B}$AR, respectively, reduces or augments the severity of cardiac hypertrophy, heart failure, and ischemic disease. In the CNS, $\alpha_{1A}$AR stimulation is antiepileptic and enhances neurogenesis, whereas excessive $\alpha_{1B}$AR activation is detrimental to brain function. In addition, $\alpha_1$ARs contribute to the regulation of immune cells[10,11], and pre-clinical studies suggest that $\alpha_1$AR-blockers may protect against hyperinflammatory responses[11–13]. The $\alpha_1$AR antagonist prazosin is currently under evaluation in clinical trials for the prevention of cytokine storm syndrome caused by the severe acute respiratory syndrome coronavirus 2 (SARS-CoV-2), a leading cause of morbidity and mortality in coronavirus disease 2019 (COVID-19; https://clinicaltrials.gov/ct2/show/NCT04365257)[12]. Similar to $\alpha_1$ARs, individual $\alpha_2$AR subtypes differentially regulate important physiological processes, such as blood pressure homeostasis, placenta development, mood, and pain perception[6,14].

The dearth of fully selective ligands for individual $\alpha$AR subtypes, particularly for $\alpha_{1B}$AR, aggravates the scarcity of brain-permeable $\alpha$AR agents to treat neurological and neuropsychiatric disorders[3,7–9,15]. The discovery of subtype-selective compounds is exceptionally challenging due to the high sequence and structural conservation in the orthosteric ligand-binding sites of closely related subtypes, which recognize the same endogenous agonists. In addition, ligand promiscuity often extends to more distant receptor subfamilies because aminergic GPCRs share several key features for ligand recognition[16,17]. For instance, the widely prescribed antipsychotic risperidone and the antidepressant amitriptyline have a high affinity for many aminergic GPCRs, including $\alpha_1$ARs[18]. Unfortunately, the undesired interaction with $\alpha_1$ARs can result in postural hypotension and related complications as side effects.

Recent high-resolution structures of aminergic GPCRs and a large body of mutagenesis studies have revealed that subtype selectivity can be achieved by exploiting secondary binding pockets, which are less conserved than the orthosteric site[16,19–27]. However, while structures of $\beta$ARs have been determined bound to various ligands[28–34], and structures of $\alpha_2$ARs recently became available[26,35,36], no $\alpha_1$AR structure has been reported to date.

In this work, to gain insights into the structure of $\alpha_1$ARs and identify key determinants of ligand selectivity, we determine the crystal structure of human $\alpha_{1B}$AR in complex with (+)-cyclazosin[37–39]. This inverse agonist shares its piperazinyl quinazoline scaffold with a series of close analogs clinically used as antihypertensive drugs[4,5] (Supplementary Fig. 1), such as prazosin, doxazosin, and terazosin, which are also currently evaluated in clinical studies to prevent a cytokine storm in COVID-19 patients (https://clinicaltrials.gov/ct2/show/NCT04365257). Remarkably, the piperazinyl quinazoline scaffold is structurally distinct from any other aminergic ligand co-crystallized so far[27]. By comparison with the sequences and structures of $\alpha_2$ARs and by pharmacological characterization of $\alpha_{1B}$AR-$\alpha_{2C}$AR chimeras with antagonists and inverse agonists with different selectivity profiles, we identify molecular determinants of selectivity within $\alpha$ARs at positions 3.29 in TM3 and 6.55 in TM6 (Ballesteros–Weinstein numbering[40]). Together, our structural and pharmacological analysis provides a basis for the design of novel drugs selectively targeting $\alpha$AR subtypes, as well as drugs devoid of detrimental off-target interactions with these receptors. Furthermore, the $\alpha_{1B}$AR structure presented here may assist the design of fully $\alpha_{1B}$AR-selective ligands to improve our understanding of this $\alpha_1$AR subtype's biological roles and explore this receptor as a therapeutic target to treat cardiovascular, neurological, and inflammatory diseases.

## Results

**Crystallization and structure determination of $\alpha_{1B}$AR.** As attempts to obtain well-diffracting crystals of a previously stabilized $\alpha_{1B}$AR mutant, $\alpha_{1B}$AR-#12[41], were unsuccessful, we selected a more stable variant using directed evolution[42] (Supplementary Table 1). To increase the chances of crystallization, we deleted the N-terminal residues M1–N34 as well as residues K249–L283 in the third intracellular loop (ICL3), and we fused the designed ankyrin repeat protein (DARPin) D12 crystallization chaperone[43] to the C-terminal end of transmembrane helix 7 (TM7) of the stabilized $\alpha_{1B}$AR variant, generating $\alpha_{1B}$AR$_{XTAL}$ (Supplementary Fig. 2). We have recently established the fusion of DARPin D12 to TM7 of a GPCR as a tool to facilitate GPCR crystallization[44]. The stabilizing mutations locked the receptor in a signaling-inactive state, as evidenced by the lack of agonist-induced $G_q$ signaling compared to wild-type $\alpha_{1B}$AR (Supplementary Fig. 3a). We observed that the following individual mutations substantially impair agonist-induced $G_q$ signaling: S95$^{2.54}$→C, S150$^{34.50}$→Y, G183$^{4.63}$→V, D191$^{ECL2}$→Y, T295$^{6.36}$→M, V333$^{7.38}$→L, F334$^{7.39}$→L, and P349$^{7.54}$→L (Supplementary Fig. 3b, c and Supplementary Table 2). We expressed $\alpha_{1B}$AR$_{XTAL}$ in the inner membrane of E. coli and isolated properly folded receptors via a prazosin ligand-affinity column (see "Methods").

The inverse agonist cyclazosin is a racemic mixture of the (+)- and (−)-enantiomers (Fig. 1a). We use the term cyclazosin to refer to this racemic mixture, and we specify which particular enantiomer where required. The affinity of cyclazosin for $\alpha_{1B}$AR$_{XTAL}$ is high and only marginally reduced compared to wild-type $\alpha_{1B}$AR ($K_i = 6.17\,\text{nM}$ and $1.02\,\text{nM}$, respectively) (Supplementary Fig. 3d, e). The fusion of DARPin D12 did not significantly perturb cyclazosin affinity. Cyclazosin-bound $\alpha_{1B}$AR$_{XTAL}$ exhibited high thermostability in a CPM assay[45] (apparent $T_m \approx 71\,°C$) (Supplementary Fig. 3f). This complex was thus suitable for crystallization in the lipidic cubic phase (LCP).

We co-crystallized $\alpha_{1B}$AR$_{XTAL}$ with cyclazosin in LCP and determined its structure (Fig. 1b, Supplementary Figs. 2b and 4, and Supplementary Table 3). Crystallization of $\alpha_{1B}$AR$_{XTAL}$ in the presence of cyclazosin yielded only a limited number of useful crystals. One hundred and nine partial datasets were recorded, carefully inspected, and bad data regions removed (see "Methods"). Data reduction of the best 27 morphologically close data wedges (between 15° and 32°), selected for the similarity in their unit cell parameters, resulted in a 98% complete dataset at 3.1 Å resolution. To correct for the moderate anisotropy of the data

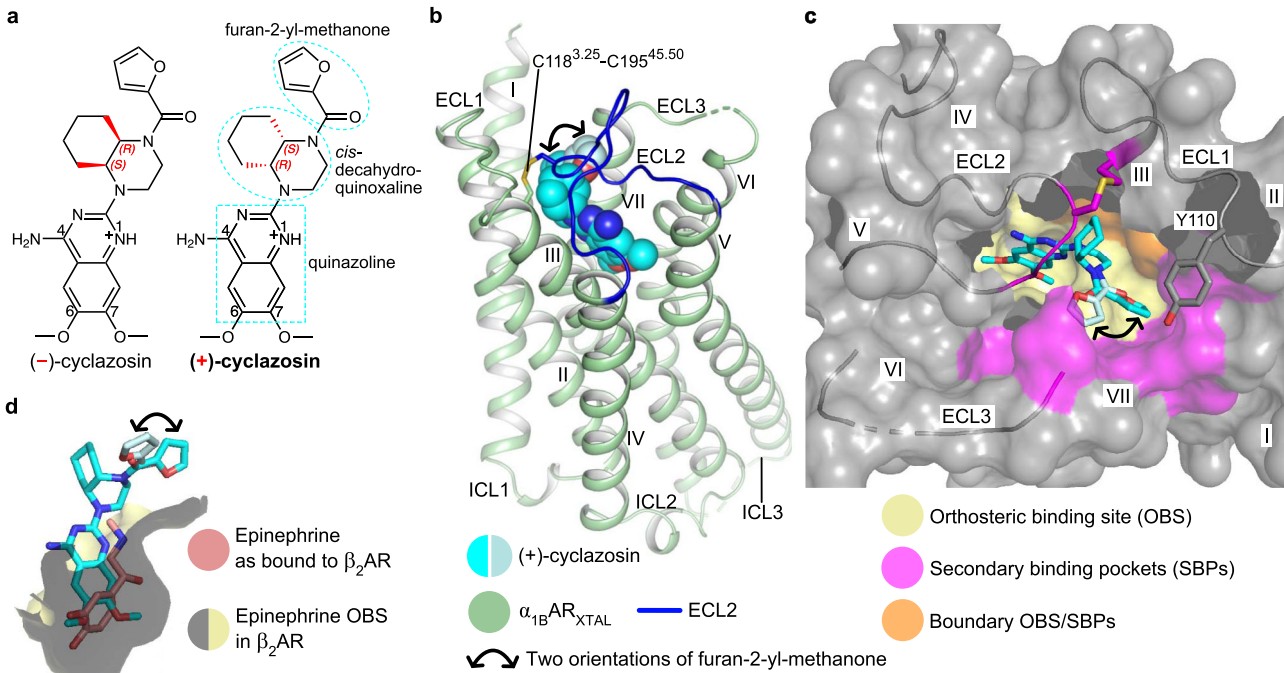

**Fig. 1 Structure of $\alpha_{1B}AR_{XTAL}$ bound to (+)-cyclazosin and overview of the ligand-binding site. a** Chemical structure of (−)- and (+)-cyclazosin. N1 is expected to be mostly protonated at the crystallization pH of 6.0 (see "Methods") as well as at physiological pH, and the resulting positive charge will be delocalized over the quinazoline ring system[49–51]. **b** Structure of $\alpha_{1B}AR_{XTAL}$ bound to (+)-cyclazosin. For clarity, DARPin D12 has been omitted. (+)-Cyclazosin is depicted as van der Waals spheres. The two orientations observed for the furan-2-yl-methanone substituent of (+)-cyclazosin are colored in cyan and pale cyan, respectively, and are indicated by a black curved arrow. Oxygen, nitrogen, and sulfur atoms are depicted in red, blue, and yellow, respectively. ECL, extracellular loop; ICL, intracellular loop. **c** Surface representation of the (+)-cyclazosin binding site in $\alpha_{1B}AR_{XTAL}$. ECL1–3 are shown as surface and as cartoon; (+)-cyclazosin is shown as sticks. The orthosteric binding site (OBS) has been approximated on the basis of the $\beta_2AR$-epinephrine complex (see panel d and main text). **d** Comparison of the binding modes of (+)-cyclazosin in $\alpha_{1B}AR_{XTAL}$ and epinephrine in $\beta_2AR$ (PDB ID: 4LDO[30]).

(deltaB ≈ 17 Å²), we submitted our unmerged data to the STARANISO server[46]. The result was an anisotropy-corrected dataset extending to 2.87 Å despite low completeness and poor data collection statistics in the highest resolution shell (Supplementary Table 3). Refinement of the cyclazosin-$\alpha_{1B}AR_{XTAL}$ complex was performed with both the 3.1-Å and 2.87-Å datasets and yielded very similar structures. However, better electron density maps and refinement statistics were obtained with the 2.87-Å anisotropy-corrected dataset. The electron density was of good quality, except for the following receptor regions: residues 35–37 of the truncated N terminus, $238^{5.73}$–$247^{ICL3}$ of the shortened ICL3, and $320^{6.61}$–$323^{ECL3}$ (Ballesteros–Weinstein numbering[40] denoted in superscript).

Strong electron density in the ligand-binding pocket allowed unambiguous modeling of (+)-cyclazosin and key receptor side chains discussed herein (Supplementary Fig. 5), except $V197^{45.52}$ in ECL2. Despite the use of racemic cyclazosin and the fact that both enantiomers have a similar high affinity for wild-type $\alpha_{1B}AR[37]$, binding of the (+)-enantiomer was favored in our crystals. Two distinct orientations of the furan-2-yl-methanone substituent of (+)-cyclazosin were observed (Fig. 1b, c and Supplementary Fig. 5a, b), which indicates a certain degree of conformational freedom for this moiety in $\alpha_{1B}AR_{XTAL}$.

**Architecture of $\alpha_{1B}AR_{XTAL}$.** $\alpha_{1B}AR_{XTAL}$ in complex with (+)-cyclazosin exhibits the canonical GPCR architecture consisting of TM1–7 connected by three intracellular loops (ICL1–3) and three extracellular loops (ECL1–3) (Fig. 1b). Helix 8 has been replaced by the DARPin D12 fusion (Supplementary Fig. 2). The ECLs lack regular secondary structure (Fig. 1b). ECL2 is

tethered to the extracellular tip of TM3 through the conserved disulfide bridge between $C195^{45.50}$ and $C118^{3.25}$, forming a partial lid on the (+)-cyclazosin binding site (Fig. 1b, c). We note, however, that crystal contacts are formed by ECL2 (Supplementary Fig. 4d).

Except for TM1, which is tilted outwards as a consequence of crystal packing (Supplementary Fig. 4d, e), the seven-transmembrane (7TM) bundle of $\alpha_{1B}AR_{XTAL}$ adopts a similar arrangement as observed in other antagonist-bound ARs, including $\alpha_{2A}AR$ and $\alpha_{2C}AR$ bound to the antagonist RS79948 (PDB IDs: 6KUX[35] and 6KUW[26], respectively) and $\beta_1AR$ and $\beta_2AR$ bound to the antagonist carazolol (PDB IDs: 2YCW[32] and 2RH1[28], respectively) (Supplementary Figs. 6 and 7). $\alpha_{1B}AR_{XTAL}$ apparently captured an inactive state, as evidenced by the closed intracellular arrangement of TM6 (Supplementary Fig. 6) and the negative activation index ($A^{100} = -45.7$)[47]. The nearly identical arrangement of the cytoplasmic end of TM7 in $\alpha_{1B}AR_{XTAL}$ compared to the other inactive-state AR structures suggests that the DARPin D12 fusion did not perturb its conformation. The extracellular end of TM3 in $\alpha_{1B}AR_{XTAL}$ is more inward-pointing compared to the above-mentioned $\alpha_2AR$ and $\beta AR$ structures (Supplementary Fig. 7). TM4 is over one helical turn longer in $\alpha_{1B}AR_{XTAL}$ compared to the $\alpha_2AR$ structures, and thus, it resembles the $\beta AR$ structures (Supplementary Fig. 7). The shorter TM4 observed in the $\alpha_2AR$ structures is likely the consequence of two consecutive proline residues at its extracellular end. The lack of regular secondary structure in ECL2 of $\alpha_{1B}AR_{XTAL}$ resembles the $\alpha_2AR$ structures, whereas in $\beta ARs$ ECL2 forms a short α-helix. Of note, $K331^{7.36}$ establishes a salt bridge with $E106^{2.65}$ in $\alpha_{1B}AR_{XTAL}$ (Supplementary Fig. 7a). Mutations abolishing the positive charge of $K331^{7.36}$ resulted in constitutive receptor

activity[48]; however, whether the K331[7.36]-E106[2.65] salt bridge is involved in receptor activation requires further studies.

**(+)-Cyclazosin extends from the orthosteric binding site toward secondary binding pockets.** (+)-Cyclazosin has a reported 100–1,000-fold selectivity for α$_1$ARs over α$_2$ARs, and a slight preference for α$_{1B}$AR over the other α$_1$AR subtypes[37]. To identify the determinants of ligand selectivity and to guide compound optimization, we first inspected the binding mode of (+)-cyclazosin in α$_{1B}$AR$_{XTAL}$, followed by sequence and structural comparisons with the closely related α$_2$ARs.

(+)-Cyclazosin adopts an inverted L-shaped binding mode in α$_{1B}$AR$_{XTAL}$ (Fig. 1c). The dimethoxyquinazoline moiety inserts deeply into the binding pocket, occupying the orthosteric binding site (OBS), i.e., the site that accommodates the endogenous agonists epinephrine and norepinephrine. As there is currently no structure of an αAR in complex with an endogenous agonist, the OBS was approximated on the basis of the β$_2$AR-epinephrine complex (Fig. 1d) (PDB ID: 4LDO[30]), in agreement with mutagenesis studies defining a common OBS within ARs[4]. The *cis*-decahydroquinoxaline moiety of (+)-cyclazosin, composed of a piperazine and a fused cyclohexane ring (Fig. 1a), is accommodated at the boundary between the OBS and a secondary binding pocket defined by TM3 and ECL2 (Fig. 1c). Finally, the two distinct conformations adopted by the furan-2-yl-methanone moiety are accommodated in secondary binding pockets proximal to the extracellular surface (Fig. 1c). (+)-Cyclazosin can be thus considered a "bitopic" ligand that fills simultaneously both the OBS and secondary binding pockets.

Twenty-five residues delineate the ligand-binding pocket of α$_{1B}$AR$_{XTAL}$ within 5 Å of (+)-cyclazosin (Fig. 2a). One of these residues is the stabilizing mutation F334[7.39]→L, which is adjacent to the V333[7.38]→L mutation. Upon back-mutation of L334[7.39] to the wild-type phenylalanine, we observed a loss of thermostability (Supplementary Fig. 3f and ref. [41]), and no crystals could be obtained despite extensive efforts. The back-mutations L334[7.39]→F and L333[7.38]→V were thus modeled into the structure of α$_{1B}$AR$_{XTAL}$, and molecular dynamics (MD) simulations were carried out on both this model as well as on the crystal structure (i.e., without back-mutations) in a lipid bilayer. Throughout these simulations, the receptor and the ligand exhibited structural stability, as evidenced by the root-mean-square deviation (RMSD) of the protein backbone (especially TM2–7, containing nearly all ligand-interacting residues) and ligand heavy atoms, respectively (Supplementary Fig. 8a, b). The simulations confirmed the ligand-binding mode observed in the crystal structure (Fig. 2b and Supplementary Fig. 8). F334$_{MD}$[7.39] favored a gauche minus (g−) χ$_1$ rotameric state (Fig. 2b) and resulted in stable binding of (+)-cyclazosin through aromatic contacts with the quinazoline and furan rings (Supplementary Fig. 8c, d). The L334[7.39]→F back-mutation did not perturb the conformation of the Y338[7.43] side chain or the hydrogen bond between the latter side chain and D125[3.32] observed in α$_{1B}$AR$_{XTAL}$ (Fig. 2a, b and Supplementary Fig. 8e), which stabilizes the binding pocket[16], nor did it affect the furan moiety of (+)-cyclazosin (Supplementary Fig. 8f, g).

The side chain of D125[3.32] in α$_{1B}$AR$_{XTAL}$ forms a charge-reinforced hydrogen bond with N1 of (+)-cyclazosin (Fig. 2a, c), which is expected to be protonated and the resulting positive charge delocalized over the quinazoline ring system[49–51] (ligand-receptor interactions are summarized in Supplementary Table 4). On one side of the pocket, the dimethoxyquinazoline moiety faces a hydrophobic patch in TM6 and TM7, consisting of W307[6.48], F310[6.51], F311[6.52], L314[6.55], and F334$_{MD}$[7.39] (Fig. 2a–c). On the opposite side of the pocket, the quinazoline ring establishes van

der Waals contacts with A122[3.29] and V126[3.33]. The cyclohexane ring of the *cis*-decahydroquinoxaline moiety protrudes toward TM3 and ECL2, and establishes van der Waals contacts with W121[3.28] and A122[3.29]. In one of the two modeled orientations, the furan ring inserts in a hydrophobic secondary pocket between TM2, TM3, and TM7, establishing van der Waals contacts with L105[2.64], W121[3.28], W335[7.40], and F334$_{MD}$[7.39], complemented by aromatic interactions with the latter three residues (Fig. 2a–c). In the alternative orientation, the furan ring points toward D327[7.32], F330[7.35], and V197[45.52] (Fig. 2a).

Overall, the dimethoxyquinazoline moiety of (+)-cyclazosin is tightly anchored within the conserved epinephrine/norepinephrine OBS, while the remaining parts of this ligand are accommodated in distinct secondary binding pockets (SBPs) or at the boundary between OBS and SBPs (Fig. 1c). The sub-pockets shaping the OBS and SBPs may offer opportunities for ligand optimization; however, a detailed understanding of their role in ligand recognition among closely related receptors is required, which we discuss next.

**Comparison of ligand-binding pockets between α$_1$ARs and α$_2$ARs.** As mentioned above, (+)-cyclazosin preferentially binds to α$_1$ARs over α$_2$ARs[37]. In contrast, the antagonist RS79948 exhibits a remarkable ~10,000-fold selectivity for α$_2$ARs over α$_1$ARs[18,52,53]. To identify key determinants of ligand selectivity in αARs, we compared the binding sites of (+)-cyclazosin in α$_{1B}$AR$_{XTAL}$ and RS79948 in α$_{2C}$AR (PDB ID: 6KUW[26]).

This comparison revealed seven non-conserved residues between α$_{1B}$AR and α$_{2C}$AR (Fig. 3a), which correspond to positions 2.64, 3.28, 3.29, 45.52, 5.43, 6.55, and 7.32. Positions 45.52 and 5.43 are only partially non-conserved among all six αARs (Fig. 3a). We then superposed the structure of α$_{1B}$AR$_{XTAL}$ bound to (+)-cyclazosin with the α$_{2C}$AR-RS79948 complex and inspected the above-mentioned non-conserved residues (Fig. 3b) (superposition of the conserved residues within the ligand-binding sites is shown in Supplementary Fig. 9). Four of the non-conserved residues form direct interactions with (+)-cyclazosin in α$_{1B}$AR$_{XTAL}$, and these residues are L105[2.64], W121[3.28], A122[3.29], and L314[6.55]. Similarly, four of the non-conserved residues form direct interactions with RS79948 in α$_{2C}$AR, and these residues are Y127[3.28], L128[3.29], L204[45.52], and Y402[6.55]. To assess the impact of these residues on selective ligand recognition, we made chimeric α$_{1B}$AR-α$_{2C}$AR mutants. For this purpose, we converted the residues in α$_{1B}$AR at the above-mentioned positions to the corresponding α$_{2C}$AR residues, either individually or in combination, and we assessed the effect on ligand affinities. We refer to the chimeric α$_{1B}$AR-α$_{2C}$AR mutants with the term "α$_{1B}$AR-α$_{2C}$" and specify in parentheses the chimeric modification. For example, α$_{1B}$AR-α$_{2C}$(L[3.29]) corresponds to α$_{1B}$AR bearing the A122[3.29]→L chimeric substitution.

**Molecular determinants of selective ligand binding to α$_2$ARs over α$_1$ARs.** We started our analysis with RS79948, as it exhibits a higher selectivity ratio [α$_2$ : α$_1$ARs ≈ 10,000 (refs. [18,53])] than (+)-cyclazosin [α$_1$ : α$_2$ARs ≈ 100–1000 (ref. [37])]. We compared the affinity of RS79948 for the chimeric α$_{1B}$AR-α$_{2C}$ mutants to that of wild-type α$_{1B}$AR and α$_{2C}$AR (Fig. 4a, Supplementary Fig. 10, and Supplementary Table 6). In agreement with previous studies[18,53], RS79948 had sub-nM affinity for α$_{2C}$AR and only μM affinity for α$_{1B}$AR. Strikingly, residue 3.29 was found to be key for RS79948 binding, as the A122[3.29]→L chimeric substitution in α$_{1B}$AR improved the affinity of this ligand by a remarkable 140-fold. L314[6.55]→Y and V197[45.52]→L resulted in 6- and 2-fold higher affinity, respectively, whereas the apparent slight improvement observed for W121[3.28]→Y did not reach statistical

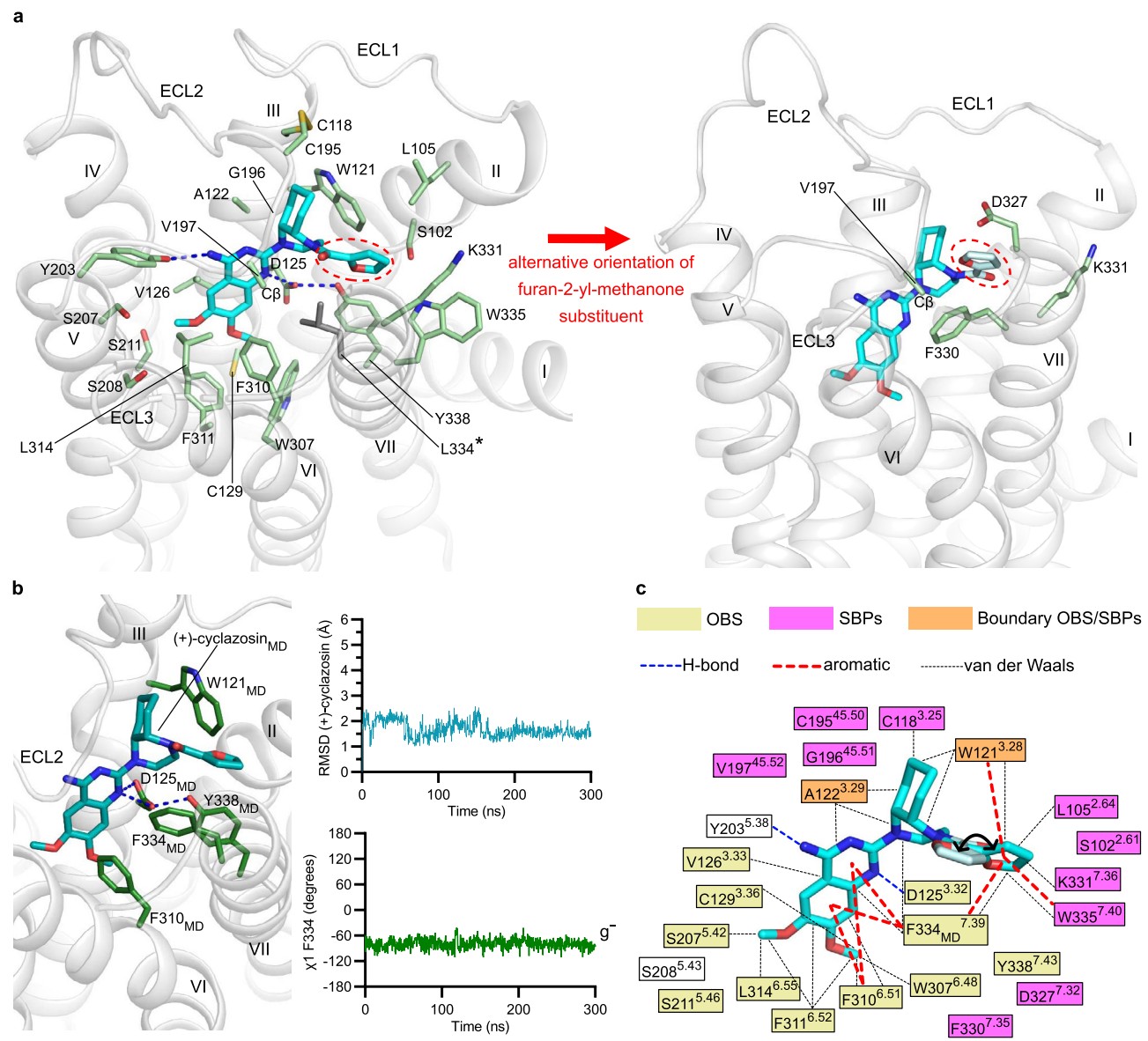

**Fig. 2 (+)-Cyclazosin binding pocket in $\alpha_{1B}AR_{XTAL}$. a** Detailed view of the (+)-cyclazosin binding site. (+)-Cyclazosin is shown as sticks in cyan, with the two alternative orientations observed for the furan-2-yl-methanone substituent (highlighted by a dashed red ellipse) colored in cyan (on the left) and pale cyan (on the right), respectively. Receptor residues are shown as sticks in pale green except for the F334→L mutation, which is colored in dark gray and is indicated by an asterisk. V197 is shown to Cβ only because its side chain is not resolved in the electron density map. Hydrogen bonds are depicted as dashed blue lines. Oxygen, nitrogen, and sulfur atoms are depicted in red, blue, and yellow, respectively. **b** MD simulation of $\alpha_{1B}AR_{XTAL-MD}$-V333-F334. The plots on the right indicate the structural stability of (+)-cyclazosin and $F334_{MD}$ throughout the simulation. RMSD, root-mean-square deviation; $g^-$, gauche minus conformation of the $\chi_1$ dihedral angle. For (+)-cyclazosin, RMSD values were calculated on all atoms. A representative snapshot of the final nanosecond of the simulation is depicted on the left, viewed from the same perspective as in panel a. (+)-Cyclazosin is colored in teal; $F334_{MD}$ is colored in dark green. **c** Schematic representation of the (+)-cyclazosin binding site. OBS, orthosteric binding site; SBPs, secondary binding pockets. A black curved arrow indicates the two orientations observed for the furan-2-yl-methanone moiety. Note that residues C195^45.50, G196^45.51, and V197^45.52 belong to ECL2, which forms crystal contacts. Source data are provided as a Source Data file.

significance (Supplementary Table 7). The remarkable impact of A122^3.29→L in $\alpha_{1B}AR$ is consistent with the observation that the reciprocal L128^3.29→A chimeric substitution in $\alpha_{2C}AR$ decreases the ability of RS79948 to antagonize agonist-induced signaling by 50-fold[26]. We also made an $\alpha_{1B}AR$ mutant bearing all four chimeric substitutions, i.e., W121^3.28→Y, A122^3.29→L, V197^45.52→L, and L314^6.55→Y — termed $\alpha_{1B}AR$-$\alpha_{2C}$(YLLY). The affinity of RS79948 for $\alpha_{1B}AR$-$\alpha_{2C}$(YLLY) was improved by almost three orders of magnitude compared to wild-type $\alpha_{1B}AR$ (Fig. 4a), approaching the affinity for $\alpha_{2C}AR$. Compared to the A122^3.29→L chimeric substitution alone, the additional gain of

affinity observed for $\alpha_{1B}AR$-$\alpha_{2C}$(YLLY) corresponded to an ~5-fold improvement. No further improvement in the affinity of RS79948 was observed by introducing chimeric substitutions at the remaining three non-conserved positions within the binding site of $\alpha_{1B}AR$, i.e., L105^2.64→N, S208^5.43→C, and D327^7.32→G (Supplementary Table 6). We also replaced the entire ECL2 in $\alpha_{1B}AR$-$\alpha_{2C}$(YLLY) with the corresponding $\alpha_{2C}AR$ sequence (from I178^4.58 to F202^5.37, i.e., including the tips of TM4 and TM5), generating $\alpha_{1B}AR$-$\alpha_{2C}$(YLLY;ECL2). However, no further improvement in RS79948 affinity was observed upon the replacement of ECL2 (Supplementary Table 6).

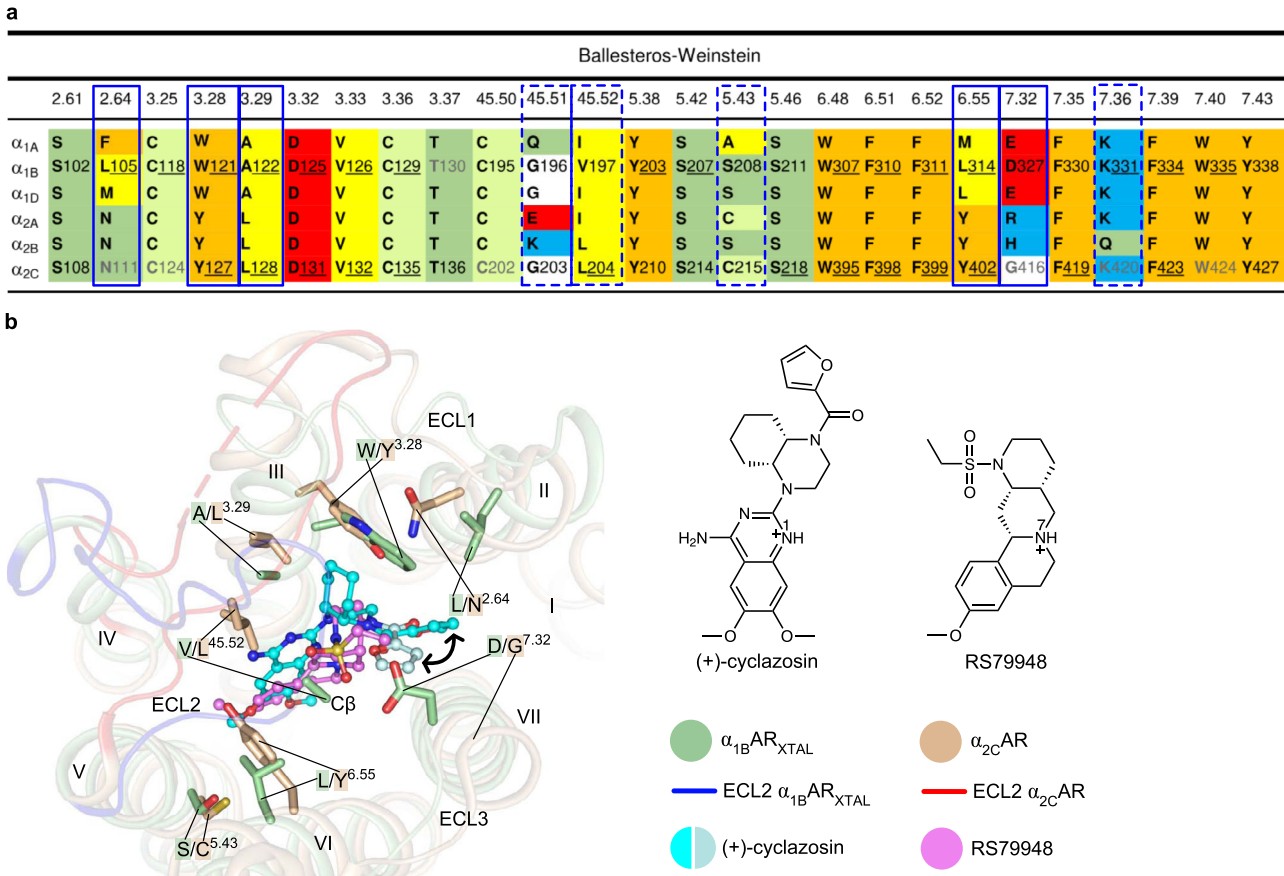

**Fig. 3 Comparison of the ligand-binding pockets of α1BAR_XTAL bound to (+)-cyclazosin and α2CAR-RS79948. a** Alignment of residues delineating the binding pockets of (+)-cyclazosin in α1BAR_XTAL and of RS79948 in α2CAR (PDB ID: 6KUW[26]). Non-conserved residues between human α1ARs and α2ARs are highlighted by solid blue rectangles, whereas dashed blue rectangles highlight partially non-conserved residues. Underlined black residues interact with the cognate ligand, whereas non-underlined black residues do not, but are nonetheless within 5 Å of it (ligand-receptor interactions are listed in Supplementary Tables 4 and 5). Gray residues are >5 Å away from the cognate ligand. Aromatic residues are highlighted in orange, hydrophobic residues in yellow, polar residues in green, Cys in yellow-green, acidic residues in red, basic residues in blue. **b** Superposition of α1BAR_XTAL bound to (+)-cyclazosin with α2CAR-RS79948, focusing on the non-conserved residues within the ligand-binding pocket (cf. panel a). Receptor residues are shown as sticks; ligands are shown in ball-and-stick representation. V197 is shown to Cβ only because its side chain is not resolved in the electron density map. A black curved arrow indicates the two orientations observed for the furan-2-yl-methanone substituent of (+)-cyclazosin. Oxygen, nitrogen, and sulfur atoms are depicted in red, blue, and yellow, respectively.

Overall, our data indicate that the presence of L128[3.29] and Y402[6.55] in α2CAR, compared to A122[3.29] and L314[6.55] in α1BAR, is the main reason for the high α2CAR selectivity of RS79948. The importance of residues 3.29 and 6.55 for the selectivity profile of RS79948 can most likely be extended to all αARs, owing to the conservation of A[3.29] within α1ARs and of L[3.29] and Y[6.55] within α2ARs (Fig. 3a). Position 6.55 deviates within α1ARs in α1AAR. In this subtype, residue 6.55 is methionine, which is, however, a large aliphatic residue as L[6.55] in α1BAR and α1DAR (Fig. 3a).

**Structural basis for α2AR-selective ligand recognition**. Inspection of the above-mentioned non-conserved residues in the structures of α1BAR_XTAL bound to (+)-cyclazosin and α2CAR bound to RS79948 (PDB ID: 6KUW[26]) provides a rationale for their role in ligand selectivity (Figs. 3b and 4b, c). In α2CAR, L128[3.29] stabilizes the position of RS79948 close to TM3 through hydrophobic contacts with the polycyclic ring system of this antagonist. This positional stabilization may favor the crucial interaction between the side chain of D131[3.32] in TM3 and the ligand's N7 atom, which is expected to be protonated (Fig. 4b). In addition, L128[3.29] stabilizes the position of L204[45.52] in ECL2 through hydrophobic contacts. In turn,

L204[45.52] restricts the binding site and establishes hydrophobic interactions with RS79948.

The corresponding residue in α1BAR, A122[3.29], may not form sufficiently strong contacts, neither with RS79948 nor with residues in ECL2 (Figs. 3b and 4c), resulting in weaker binding of this ligand. The conformation adopted by ECL2 also substantially differs between the two complexes (Figs. 3b and 4c), although this should be interpreted with caution because these structures are bound to different ligands, and crystal contacts are established by ECL2. Nonetheless, the different conformation adopted by ECL2 on top of the binding sites in α1BAR and α2CAR suggests a role for this region in ligand selectivity. As the course of ECL2 is influenced by contacts with other receptor regions, it is difficult to reproduce its effect on ligand selectivity using chimeras.

We also note an inward tilting of the extracellular end of TM3 in α1BAR_XTAL compared to the α2CAR-RS79948 complex, as well as a different rotamer adopted by the side chain of D[3.32] (Fig. 4c). The ligand-binding pocket is thus narrower in α1BAR_XTAL in this region, which may preclude RS79948 from binding deeply in the TM bundle. Thus, depending on the plasticity of this receptor region, the shape of the pocket in the proximity of D[3.32] will possibly contribute to ligand selectivity.

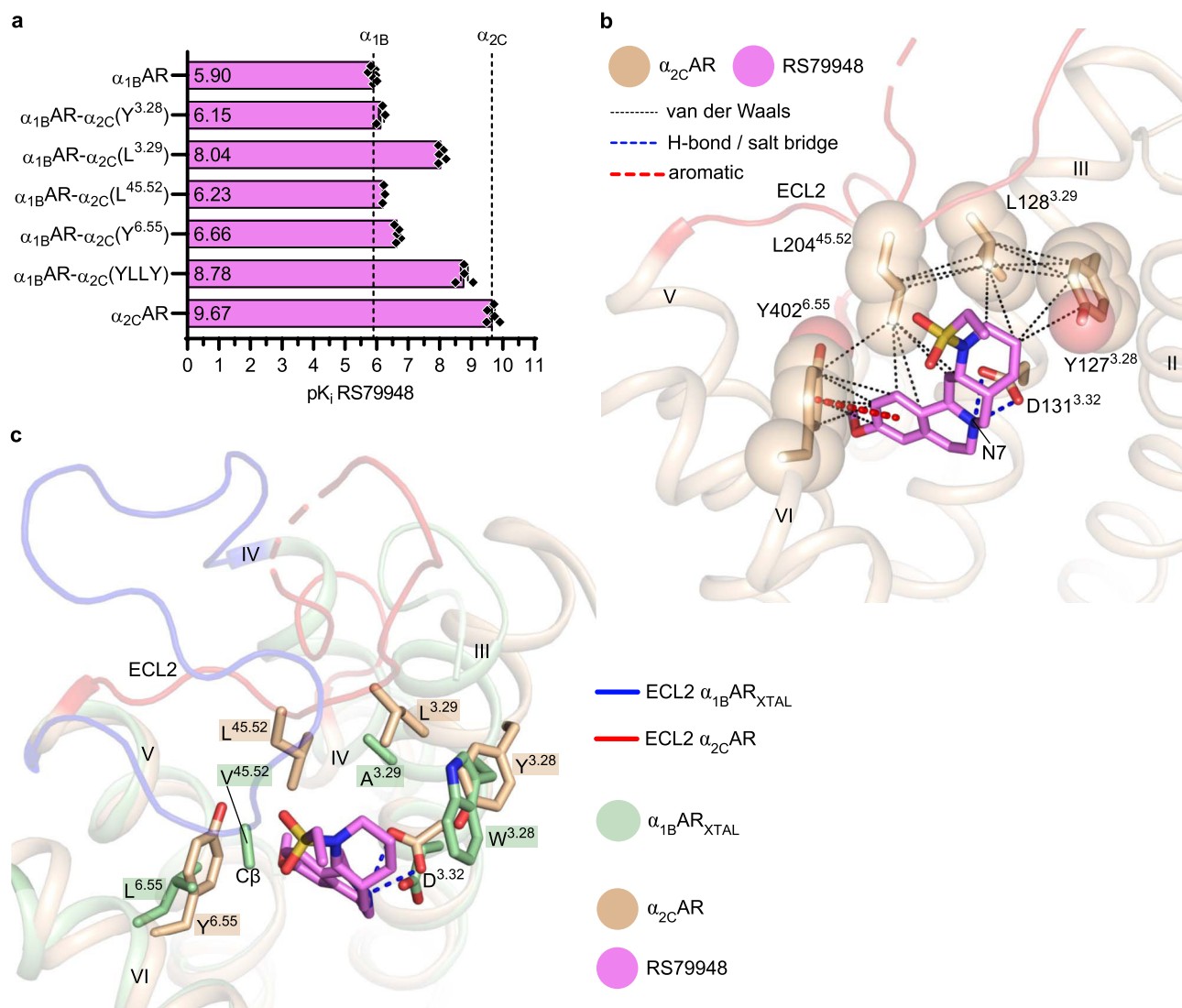

**Fig. 4 Molecular determinants and structural basis for the selective binding of RS79948 to $\alpha_{2C}$AR over $\alpha_{1B}$AR. a** Affinity of RS79948 for $\alpha_{1B}$AR, $\alpha_{2C}$AR, and chimeric $\alpha_{1B}$AR-$\alpha_{2C}$ mutants. Single amino acid substitutions in $\alpha_{1B}$AR are indicated in the construct names and correspond to the $\alpha_{2C}$AR sequence at either one of positions 3.28, 3.29, 45.52, or 6.55. The $\alpha_{1B}$AR-$\alpha_{2C}$(YLLY) chimera corresponds to the quadruple mutant. Data are shown as mean values ± standard error of the mean (SEM) of 3–6 independent experiments performed in triplicate. The underlying data points are depicted as black diamonds, and the exact n, SEM, and 95% confidence interval of the mean are reported in Supplementary Table 6. Differences in affinities were evaluated by a statistical test as detailed in Supplementary Table 7. **b** Structural role of Y127[3.28], L128[3.29], L204[45.52], and Y402[6.55] in the binding of RS79948 to $\alpha_{2C}$AR (PDB ID: 6KUW[26]). TM1, ECL3, and TM7 have been omitted for clarity. Receptor residues are shown as van der Waals spheres and as sticks, except for D131[3.32], which is shown as sticks only; RS79948 is shown as sticks. Oxygen, nitrogen, and sulfur atoms are depicted in red, blue, and yellow, respectively. **c** Superposition of $\alpha_{1B}$AR$_{XTAL}$ bound to (+)-cyclazosin and $\alpha_{2C}$AR-RS79948 (PDB ID: 6KUW[26]), focusing on the residues outlined in panel b. (+)-Cyclazosin, TM1, TM2, ECL1, ECL3, and TM7, have been omitted for clarity. Receptor residues and RS79948 are shown as sticks. V197 is shown to Cβ only because its side chain is not resolved in the electron density map. Source data are provided as a Source Data file.

On the opposite side of the RS79948-binding pocket in $\alpha_{2C}$AR, Y402[6.55] establishes aromatic and van der Waals interactions with the benzene moiety of this antagonist, probably restricting its degrees of freedom (Fig. 4b). The corresponding L314[6.55] residue in $\alpha_{1B}$AR, owing to its aliphatic side chain and smaller size, may form weaker interactions and allow more flexibility of this ligand (Figs. 3b and 4c). Due to the constrained polycyclic ring structure of RS79948, mispositioning of the benzene moiety could again negatively affect the interaction between the ligand's N7 atom and the side chain of D131[3.32].

Finally, in addition to modulating RS79948 selectivity, the different properties of A[3.29] in $\alpha_1$ARs compared to L[3.29] in $\alpha_2$ARs may underlie the distinct selectivity profiles of yohimbine and

corynanthine (also known as rauhimbine), two RS79948-related compounds (Supplementary Fig. 11a). Although yohimbine and corynanthine differ in the configuration of only a single stereocenter (C5), yohimbine is a potent $\alpha_2$AR-selective antagonist, while corynanthine has a substantially reduced affinity for $\alpha_2$ARs[54–56]. Docking of these ligands to the $\alpha_{2C}$AR structure (PDB ID: 6KUW[26]) and to a model of $\alpha_{2C}$AR-L128[3.29]→A, together with MD simulations, suggests that binding of corynanthine is hampered by steric hindrance between its bulky methyl ester substituent and L[3.29] in $\alpha_2$ARs (Supplementary Fig. 11b–g and Supplementary Movie 1). In contrast, the same methyl ester substituent in yohimbine points away from L[3.29], owing to the opposite stereochemical configuration at C5,

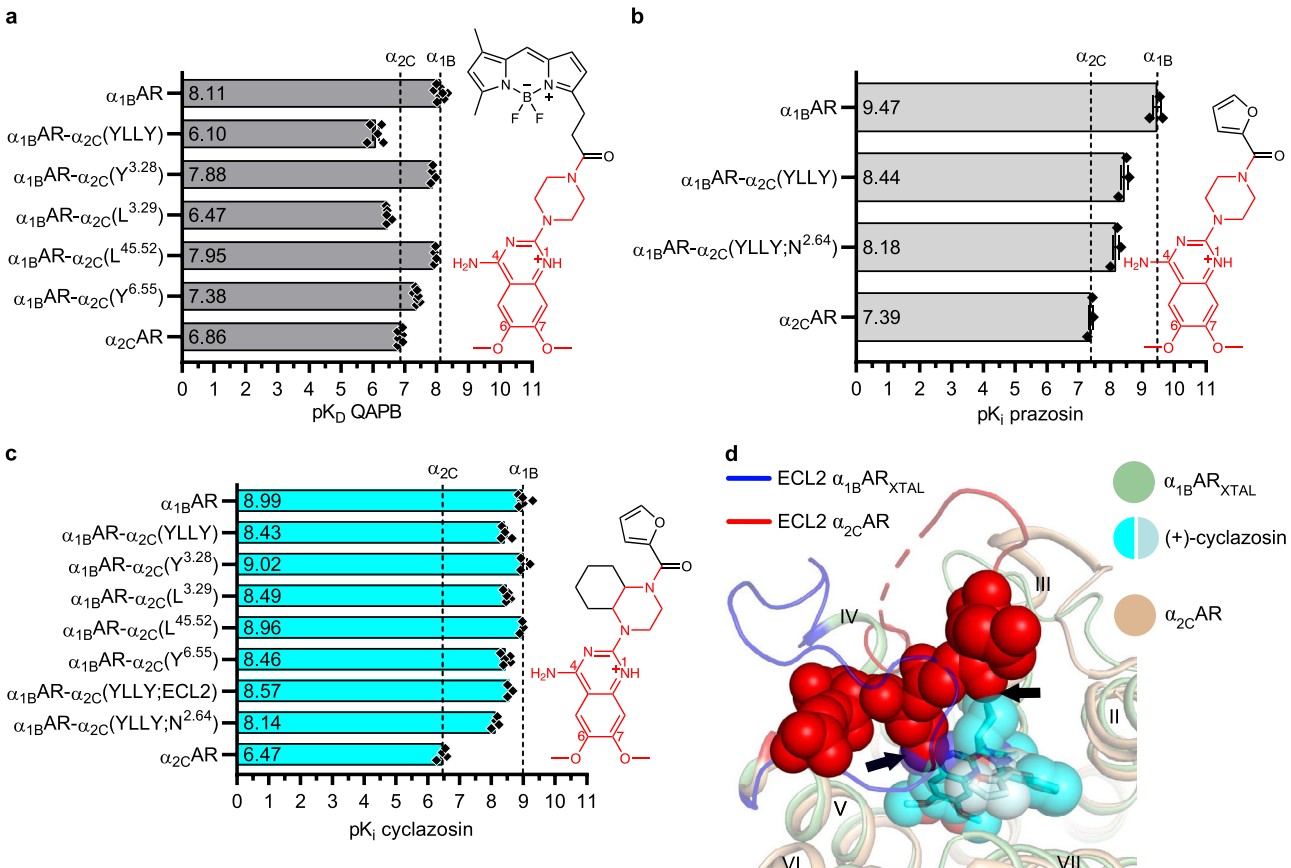

**Fig. 5 Molecular determinants for the preferred binding of piperazinyl quinazolines to α₁ARs over α₂ARs. a–c** Affinity of (**a**) QAPB, (**b**) prazosin, and (**c**) cyclazosin for α₁BAR, α₂CAR, and chimeric α₁BAR-α₂C mutants. Single amino acid substitutions in α₁BAR are indicated in the construct names and correspond to the α₂CAR sequence at either one of positions 3.28, 3.29, 45.52, or 6.55. The α₁BAR-α₂C(YLLY) chimera corresponds to the quadruple mutant. All three ligands share a common piperazinyl 4-amino-6,7-dimethoxyquinazoline scaffold, which is highlighted in red in their chemical structures. Data are shown as mean values ± SEM of 3–6 independent experiments performed in triplicate. The underlying data points are depicted as black diamonds, and the exact *n*, SEM, and 95% confidence interval of the mean are reported in Supplementary Table 6. Differences in affinities were evaluated by a statistical test as detailed in Supplementary Table 7. **d** Superposition of α₁BAR_XTAL bound to (+)-cyclazosin and α₂CAR-RS79948 (PDB ID: 6KUW[26]), focusing on the potential role of ECL2 in selective ligand binding. The ECL2 residues in α₂CAR that form a lid on the ligand-binding site are shown as red van der Waals spheres. (+)-Cyclazosin is depicted as sticks and as transparent van der Waals spheres, with the two orientations observed for its furan-2-yl-methanone substituent colored in cyan and pale cyan, respectively. Black arrows indicate potential hindrance exerted by ECL2 with respect to (+)-cyclazosin. Source data are provided as a Source Data file.

averting steric hindrance with L[3.29]. Less hindrance is expected for binding of corynanthine to α₁ARs, since α₁ARs bear an alanine at position 3.29.

**Molecular determinants of preferred ligand binding to α₁ARs over α₂ARs.** To gain further understanding of selective ligand recognition within αARs, we investigated the preferential binding of piperazinyl 4-amino-6,7-dimethoxyquinazoline compounds to α₁ARs over α₂ARs[57]. Three such compounds, i.e., QAPB (quinazolinyl piperazine BODIPY), prazosin, and cyclazosin, displayed ~20-, 120-, and 330-fold higher affinity for wild-type α₁BAR compared to α₂CAR, respectively (Fig. 5a–c, Supplementary Fig. 10, and Supplementary Table 6). The measured affinities agree well with previously reported values[15,18]. The α₁BAR-α₂C(YLLY) chimera described above had ~100-, 11-, and 4-fold weaker affinity for QAPB, prazosin, and cyclazosin, respectively, compared to wild-type α₁BAR (Fig. 5a–c). When assessed individually, the chimeric substitutions A122[3.29]→L and L314[6.55]→Y led to a 45- and 5-fold loss of QAPB affinity at α₁BAR, respectively. These two substitutions also lowered cyclazosin affinity by ~3-fold each, whereas the effect of W121[3.28]→Y and V197[45.52]→L was negligible for both QAPB and cyclazosin. Based

on these trends, it is conceivable that the higher steric bulk of L[3.29] and Y[6.55] in α₂ARs, compared to A[3.29] and L/M[6.55] in α₁ARs, hinders an optimal positioning of these piperazinyl quinazoline ligands in the binding pocket.

To assess the contribution of hydrophobic contacts mediated by the furan ring of prazosin and cyclazosin with TM2 (Fig. 2a, c) to the selectivity profile, we introduced the chimeric substitution L105[2.64]→N, generating α₁BAR-α₂C(YLLY;N[2.64]). The affinity of prazosin and cyclazosin for this chimera appeared to decrease by ~2-fold compared to α₁BAR-α₂C(YLLY) (Fig. 5b, c), however, this loss of affinity did not reach statistical significance (Supplementary Table 7). No further loss of cyclazosin affinity was observed for chimeric substitutions at the remaining two non-conserved positions (S208[5.43]→C and D327[7.32]→G) within the binding site of α₁BAR (Supplementary Table 6).

We hypothesized that ECL2 of α₂ARs may play a role in the selectivity of piperazinyl 4-amino-6,7-dimethoxyquinazoline compounds for α₁ARs over α₂ARs. Thus, we measured the affinity of cyclazosin for the α₁BAR-α₂C(YLLY;ECL2) chimera described above. No change in cyclazosin affinity was observed at α₁BAR-α₂C(YLLY;ECL2) compared to α₁BAR-α₂C(YLLY) (Fig. 5c). Nonetheless, superposition of the (+)-cyclazosin complex with

the structures of $\alpha_{2A}$AR and $\alpha_{2C}$AR bound to RS79948 suggests that ECL2 hinders binding of piperazinyl 4-amino-quinazoline ligands to $\alpha_2$ARs, in particular those with a constrained bulky group fused to the piperazine ring, such as cyclazosin (Fig. 5d and Supplementary Fig. 7a, b). It is possible, however, that the $\alpha_{1B}$AR-$\alpha_{2C}$(YLLY;ECL2) chimera could not reproduce the effect of ECL2 on ligand selectivity because the conformation of this loop in $\alpha_2$ARs is influenced by contacts with other receptor regions, as previously described[26]. The fact that the conformation of ECL2 in $\alpha_{2A}$AR has already been shown to hinder binding of bulky ligands[26] is consistent with our postulated mechanism of selectivity.

Overall, we identified positions 3.29 and 6.55 as contributors to the $\alpha_1$AR selectivity of piperazinyl quinazoline compounds. Nonetheless, the molecular basis for the $\alpha_1$AR selectivity of prazosin and cyclazosin could not yet be fully replicated from chimeras. We thus postulate that ECL2 in $\alpha_2$ARs may sterically interfere with the binding of these ligands, in particular of cyclazosin, and thus contributes to the selectivity of these ligands for $\alpha_1$ARs.

## Discussion

The emerging key roles of $\alpha$ARs in many physiological and pathophysiological processes, especially in the heart, CNS, and immune system[6–11,14], have suggested new avenues to treat cardiovascular, neurological, neuropsychiatric, and inflammatory disorders. Clearly, selective targeting of individual $\alpha$AR subtypes will be crucial to fully exploit these receptors as drug targets. Moreover, due to the complexity of CNS disorders, their therapy often requires polypharmacology (i.e., interaction with multiple targets), which needs to be perfectly tailored to avoid severe off-target side effects.

A particular challenge is the high conservation of key ligand-anchoring residues in the binding pockets of aminergic receptors[16,17]. For instance, the antipsychotics risperidone and haloperidol bind to the D2 dopamine receptor (DRD2)[20,22] through contacting residues that are nearly entirely conserved in $\alpha_1$ARs (Supplementary Fig. 12), explaining some of the off-target activity of these drugs[18]. To guide the drug discovery process—even on other targets—it is crucial to understand the molecular determinants underlying ligand selectivity or promiscuity at $\alpha$ARs.

The lack of structures for $\alpha_1$ARs had largely precluded rational ligand design so far, and fully $\alpha_{1B}$AR-selective compounds are not available yet. To bridge this gap, we determined the crystal structure of human $\alpha_{1B}$AR. To enable crystallization, we had to introduce the receptor modifications typically required for crystallization of GPCRs[58], i.e., stabilizing mutations, fusion of a crystallization chaperone, and truncation of long flexible regions. We stabilized the receptor in an inactive state using CHESS-based directed evolution[42] and fused it to the DARPin D12 crystallization chaperone[43]. The same strategy has recently allowed us to determine structures for several ligand complexes of NTSR1[44], which had been recalcitrant to crystallization before. Previously reported affinity data for the inverse agonist prazosin indicate that ligand binding to $\alpha_{1B}$AR expressed in *E. coli* is not different from $\alpha_{1B}$AR produced in mammalian cells[15,59]. In addition, we have recently shown that the ligand-binding site of the neurotensin receptor 1 (NTSR1) is virtually identical in structures obtained from bacterial or eukaryotic expression systems[44]. Although our study focuses on inactive-state $\alpha_{1B}$AR, we note that the ICL3 deletion and the replacement of helix 8 by the DARPin fusion are likely to impair coupling to intracellular signaling proteins.

Alongside the previously reported $\beta$AR and $\alpha_2$AR structures, the complex of $\alpha_{1B}$AR with (+)-cyclazosin now offers a comprehensive view of ligand binding at this clinically important subfamily of GPCRs. Previous studies had identified the non-conserved residue F/N[7.39] as a crucial determinant of $\beta$AR versus $\alpha$AR selectivity[17,60]. Our findings now reveal that residues A/L[3.29] and L/Y[6.55] are important determinants of selectivity between $\alpha_1$ARs and $\alpha_2$ARs. Among aminergic receptors, A[3.29] is unique to $\alpha_1$ARs. L[6.55] is unique to $\alpha_{1B}$AR and $\alpha_{1D}$AR, while M[6.55] as in $\alpha_{1A}$AR is only found in the histamine H3 receptor (HRH3)[16]. Thus, both positions could now be exploited to design selective compounds devoid of unwanted side effects.

In aminergic GPCRs, residue 3.29 delineates the boundary between the orthosteric binding site and secondary binding pockets (the latter are also described as "extended", "minor", or "allosteric" binding sites in other studies). Due to its gatekeeper position, residue 3.29 can be expected to mediate ligand selectivity in other aminergic GPCRs, as sequence divergence is observed at the level of aminergic subfamilies, their subgroups, and even between certain subtypes within the same subgroup[16]. Indeed, in DRD2 and DRD4, the non-conserved residue at position 3.29 contributes to ligand selectivity[61]. In the muscarinic acetylcholine receptors, W[3.28]→A and L[3.29]→A differentially affect the binding of several orthosteric ligands, depending on their extension toward the extracellular side[24,62–65].

Characterization of unique secondary binding pockets for different receptor subtypes is a promising strategy for identifying subtype-selective ligands via structure-based approaches. Since there is great interest in obtaining compounds with exquisite selectivity for $\alpha_{1B}$AR over $\alpha_{1A}$AR, we note that the $\alpha_{1B}$AR structure presented here suggests to leverage on the non-conserved cavity between TM3 and TM2 (L/F[2.64]). In analogy, in DRD2 and DRD4, sequence divergence in TM2 at and proximal to position 2.64 contributes to ligand selectivity, likely by steric effects[61]. Alternatively, one might take advantage of the $\alpha_{1B}$AR residue A[5.39], which is spatially proximal to position 6.55 and less bulky than V[5.39] in $\alpha_{1A}$AR, to modulate the ligand's interaction with D[3.32]. The conformation of the C-terminal portion of ECL2, which partially caps the (+)-cyclazosin binding site, might deviate in the $\alpha_{1A}$AR and $\alpha_{1D}$AR subtypes compared to $\alpha_{1B}$AR due to non-conserved residues at the extracellular tip of TM5 (A/P[5.36] and G/F[5.37]) as well as within the loop itself. Sequence variation within ECL2 has been shown to be a source of subtype selectivity for some $\alpha$-adrenergic ligands[26,48,66].

Finally, the binding mode of (+)-cyclazosin suggests a rationale for its inverse agonist properties. The extension of this ligand toward the extracellular vestibule in the proximity of TM7 possibly prevents F[7.39] from sealing the OBS, which is triggered by agonistic ligands, and has been proposed to contribute to receptor activation[35] (Supplementary Fig. 13).

In conclusion, this study presents the crystal structure of an $\alpha_1$AR, enabled by directed evolution and a recently established DARPin fusion-based crystallization design. We elucidated key molecular determinants of $\alpha_1$AR versus $\alpha_2$AR selectivity, improving our understanding of adrenergic GPCRs and providing new opportunities for structure-based ligand screening and rational drug design.

## Methods

**Directed evolution of $\alpha_{1B}$AR**. Error-prone PCR was applied to the enriched pool from a previously described library of $\alpha_{1B}$AR (termed EP2AS3)[41]. The resultant library, with a diversity of ~300,000 clones, was subjected to one round of bacterial display for high expression[67] using the fluorescent ligand QAPB (quinazolinyl piperazine BODIPY, also termed BODIPY FL prazosin, ThermoFisher Scientific). From the 1% most fluorescent (highest QAPB-bound) cell population, 100,000 cells were sorted by fluorescence-activated cell sorting (FACS), as previously described[41]. This resultant population was subsequently subjected to two rounds of CHESS (sorting of encapsulated *E. coli* for detergent stability of the receptor)[42]. The first round was carried out with solubilization in PBS-E (10 mM Na$_2$HPO$_4$,

1.8 mM KH$_2$PO$_4$, 137 mM NaCl, 2.7 mM KCl, 1 mM EDTA, pH 7.4) containing 2% (w/v) n-decyl-β-D-maltopyranoside (DM, Anatrace), 0.5% (w/v) CHAPS (Sigma-Aldrich), and 200 nM QAPB, for 2 h at 4 °C, followed by incubation in PBS-E containing 1% (w/v) 1,2-diheptanoyl-sn-glycero-phosphocholine (DH$_7$PC, Anatrace) and 200 nM QAPB, for 30 min at 4 °C, and then PBS-E containing 0.33% (w/v) DH$_7$PC and 200 nM QAPB, for 24 h at 25 °C. Of the highest (top 1%) QAPB fluorescent capsules, 26,000 capsules were sorted from the population using FACS, and the selected clones were isolated from the capsules with PCR, before being re-ligated into the expression vector as described previously[41]. The second round of CHESS was applied with encapsulated cells solubilized as for the first round (see above), but with a final incubation in PBS-E containing 0.33% (w/v) DH$_7$PC and 200 nM QAPB, for 24 h at 37 °C. Of the highest (top 1%) QAPB fluorescent capsules, 220,000 capsules were sorted, the clones isolated, re-ligated, and subjected to a final round of bacterial display, performed as above. Twenty-four selected α$_{1B}$AR clones were screened individually as previously described[41]. Following characterization of the clones, the best-behaved clone was referred to as α$_{1B}$AR-B1 and is described elsewhere[59]. Twelve of the 14 amino acid substitutions harbored by α$_{1B}$AR-B1 compared to wild-type human α$_{1B}$AR were transferred to the crystallization construct (see below and Supplementary Table 1), as they were in receptor regions not subjected to truncations.

**Generation of α$_{1B}$AR$_{XTAL}$.** The annotated amino acid sequence of α$_{1B}$AR$_{XTAL}$ is available in the PDB under the accession code 7B6W as well as in Supplementary Fig. 2c. Briefly, α$_{1B}$AR$_{XTAL}$ was obtained by introducing the following modifications in the sequence of wild-type human α$_{1B}$AR: introduction of 12 amino acid substitutions (Supplementary Table 1) derived from directed evolution (see above), deletion of N-terminal residues M1–N34 and intracellular loop three (ICL3) residues K249–L283, and fusion of DARPin D12[43,68,69] to residue C350[7.55] via the linker sequence AEDLVEDWE (Supplementary Fig. 2a, c). This sequence has previously been used to fuse DARPin D12 to the neurotensin receptor 1 (NTSR1)[44], resulting in a shared helix between TM7 of NTSR1 and the N-terminal region of DARPin D12; however, in the complex of α$_{1B}$AR$_{XTAL}$ with (+)-cyclazosin, this linker region lacked regular secondary structure (Supplementary Fig. 2b). DARPin D12 was modified in its N-terminal region by deletion of residues S1 and D2, and by introduction of four point mutations, i.e., L3→K, G4→A, K5→R and A13→K (Supplementary Fig. 2a, c), as a result of sequence optimization with Rosetta *fixbb*[70]. The aim was to design a fusion site between GPCR and DARPin featuring a compromise between rigidity and malleability in order to adapt to different crystal packings. Furthermore, the last two C-terminal DARPin D12 residues, L157 and N158, were mutated to alanine. As depicted in Supplementary Fig. 2c, the DARPin is followed by a short linker (sequence: TRE), followed by a cleaved human rhinovirus (HRV) 3C protease cleavage site (sequence: LEVLFQ). This turned out to be partially α-helical and established crystal contacts (Supplementary Figs. 2b and 4d). For expression in *E. coli*, the gene encoding the receptor construct was cloned into a previously described pBR322-derived vector[71]. Briefly, this resulted in an expression construct consisting of an N-terminal maltose-binding protein (MBP), followed by a His$_6$-tag, a HRV 3C protease cleavage site (sequence: LEVLFQGP), a short linker (sequence: GS), the receptor itself fused to the modified DARPin D12, a short linker (sequence: TRE), a second HRV 3C protease cleavage site (sequence: LEVLFQGP), followed by thioredoxin A (TrxA) and a C-terminal His$_{10}$-tag. The HRV 3C protease cleaves the peptide bond between Q and G of the above-mentioned cleavage site.

**Generation of the prazosin ligand-affinity column.** The prazosin column consists of a maleimide-(PEG2)$_2$-prazosin derivative (Supplementary Fig. 14) coupled to the unique C-terminal Cys added via a linker to protein D (pD-Cys), which is in turn coupled via amino groups to NHS-activated Sepharose beads. pD-Cys corresponds to the pD-NT variant previously described[71], with the difference that the C-terminal HRV 3C protease cleavage site and the NT8–13 epitope are replaced by the amino acid sequence GGGGSGGGC. Expression and purification of pD-Cys were carried out as described for pD-NT[71], with the difference that EDTA pH 8.0 was added to a final concentration of 10 mM to the protein sample after elution from the Ni-NTA column, followed by protein concentration and dialysis as described[71]. Subsequently, to protect the cysteine from coupling to the NHS-activated Sepharose (otherwise a side reaction we suspected to occur), the pD-Cys protein was dimerized via a disulfide bond. To this end, CuCl$_2$ was added to a final concentration of 1 mM and the mixture incubated overnight at 4 °C, followed by 1 h at room temperature. Typically, ~85% of pD-Cys could be dimerized according to analytical gel filtration on an S200 S/150 GL column. Coupling of dimerized protein via its lysines and N terminus to NHS-activated Sepharose (GE Healthcare) was carried out as described[71], followed by blocking with Tris, but the subsequent washing buffers were adjusted to contain 10 mM EDTA pH 8.0. After washing with guanidine hydrochloride and H$_2$O as described[71], the disulfide bond was reduced by incubation with two column volumes (CV) of Reducing Buffer (50 mM HEPES pH 8.0, 25 mM TCEP, 10 mM EDTA pH 8.0) for 30 min at room temperature. Afterward, the resin was washed with Coupling Buffer (100 mM Na phosphate pH 6.0, degassed) and subsequently incubated with 1 CV of 0.8 mM maleimide-(PEG2)$_2$-prazosin (Anawa) in Coupling Buffer for 30 min at room temperature. This resulted in ~0.5 nmol ligand/μl resin as measured by comparison of the prazosin absorption at 330 nm before and after coupling. The resulting prazosin

affinity column was finally washed with Coupling Buffer and H$_2$O, and stored in 20% EtOH at 4 °C. According to our experience, the prazosin column can be used multiple times and is stable for years without any major loss in binding capacity.

**Expression and purification of α$_{1B}$AR$_{XTAL}$.** α$_{1B}$AR$_{XTAL}$ was expressed in *E. coli* BL21 cells bearing a deletion of the *fhuA2* gene to confer phage T1 resistance (New England Biolabs) as previously described for other stabilized α$_{1B}$AR variants[59]. The resulting cell pellet was resuspended with Resuspension Buffer (100 mM HEPES pH 8.0, 30% (v/v) glycerol, 400 mM NaCl) at 4 °C, frozen in liquid nitrogen, and stored at −80 °C. All the following steps were carried out at 4 °C. Typically, 90 ml of frozen resuspended cells (corresponding to 30 g of the pellet) were thawed, 45 ml of H$_2$O were added, and the resuspension was incubated with 2 mg/ml lysozyme (Sigma-Aldrich), 5 mM MgCl$_2$, and 0.05 mg/ml DNase I (Roche) for 10 min while stirring. Subsequently, receptors were solubilized by incubation with 1.67% (w/v) n-dodecyl-β-D-maltopyranoside (DDM, Anatrace) and 0.33% (w/v) cholesteryl hemisuccinate (CHS, Sigma-Aldrich) for 30 min while stirring followed by sonication for 15 min using a Sonifier 250 (Branson) at a duty cycle of 30% and output 5. The lysate containing detergent-solubilized receptors was stirred for another 1 h, then adjusted with imidazole (pH 8.0) to a final concentration of 15 mM and centrifuged at 20,000 × *g* for 30 min. The supernatant was batch-incubated for 2.5 h with 30 ml of TALON Superflow resin (GE Healthcare) equilibrated with TALON Wash Buffer I (TWB-I; 25 mM HEPES pH 8.0, 10% (v/v) glycerol, 600 mM NaCl, 0.1% (w/v) DDM, 0.02% (w/v) CHS, 15 mM imidazole pH 8.0). Subsequently, the resin was washed with 18 column volumes (CV) of TWB-I followed by 18 CV of TWB-II (25 mM HEPES pH 7.0, 10% (v/v) glycerol, 150 mM NaCl, 0.1% (w/v) DDM, 0.02% (w/v) CHS). Protein elution was carried out with 3 CV of TALON Elution Buffer (TEB; 25 mM HEPES pH 8.0, 10% (v/v) glycerol, 150 mM NaCl, 0.1% (w/v) DDM, 0.02% (w/v) CHS, 200 mM EDTA pH 8.0).

The eluted protein was concentrated in six 100 kDa molecular weight cutoff Vivaspin 20 concentrators (Sartorius) to a total volume of ~30 ml. Subsequently, the protein was incubated for 2 h with HRV 3C protease (produced in-house) to cleave off the fusion proteins MBP and TrxA, after which 2 ml of prazosin column (PC) resin equilibrated in TEB was added and incubated overnight while rolling in a Falcon tube. The resin was subsequently transferred into an empty PD-10 column, washed with 2 CV of TEB, four times with 3 CV of PCWB-I (25 mM HEPES pH 8.0, 10% (v/v) glycerol, 600 mM NaCl, 0.05% (w/v) DDM, 0.01% (w/v) CHS, 200 mM EDTA pH 8.0), three times with 2 CV of PCWB-II (10 mM HEPES pH 7.5, 10% (v/v) glycerol, 100 mM NaCl, 0.018% (w/v) DDM, 0.0036% (w/v) CHS), and partially equilibrated with 0.3 CV of PCEB (10 mM HEPES pH 7.5, 10% (v/v) glycerol, 100 mM NaCl, 0.018% (w/v) DDM, 0.0036% (w/v) CHS, 75 μM cyclazosin). For elution of cyclazosin-bound receptor, 0.75 CV of PCEB were added to the column and incubated while rolling for 2.5 h. After the addition of an additional 30 μM cyclazosin (Sigma-Aldrich), the eluant as well as a wash with 0.5 CV of PCEB were collected, resulting in an elution volume of 2–2.5 ml. In order to determine the protein concentration, excess cyclazosin had to be removed. To this end, an analytical fraction of the sample was loaded onto a Zeba Spin desalting column (7 kDa molecular weight cutoff, ThermoFisher Scientific) equilibrated with PCWB-II. Absorption at 280 nm was measured using a Nanodrop 2000 spectrophotometer (ThermoFisher Scientific) and it was corrected by multiplication with an empirically determined correction factor of 0.7 to take into account absorption of receptor-bound cyclazosin. A suitable fraction of the protein sample was then concentrated with a 100 kDa molecular weight cutoff Vivaspin 2 concentrator (Sartorius) to ~50 mg/ml, resulting in a typical final volume of 25 μl.

**Crystallization of α$_{1B}$AR$_{XTAL}$ in the lipidic cubic phase.** Cyclazosin-bound α$_{1B}$AR$_{XTAL}$ was reconstituted in lipidic cubic phase (LCP) by mixing concentrated protein (~50 mg/ml) with molten monoolein (Sigma-Aldrich) supplemented with 10% (w/w) cholesterol (Sigma-Aldrich) at a protein-to-lipid ratio of 20:31 (v/v) using the two-syringe method (100-μl syringes, Hamilton). Crystallization trials were carried out at 20 °C in 96-well glass sandwich plates (SWISSCI) with a 120-μm spacer. A Crystal Gryphon LCP crystallization robot (Art Robbins Instruments) was used to dispense either 25 nl or 40 nl boli and to cover them with 800 nl of precipitant solution. The plates were immediately sealed with a cover glass and incubated at 20 °C in a Rock Imager 1000 (Formulatrix). The crystals obtained in this study were of rather small size, typically not exceeding ~15–30 μm in any dimension. Optimized crystallization conditions consisted of 100 mM MES pH 6.0, 400–480 mM Li citrate, 28–33% (w/v) PEG400, 10 mM L-glutathione reduced form, 10 μM cyclazosin. 25 nl LCP boli tended to yield better-diffracting crystals. Crystals were harvested by picking the entire bolus at room temperature with a 25-μm MicroMesh (MiTeGen) and flash-frozen in liquid nitrogen without adding further cryoprotectant.

**Data collection, structure determination, and structural analysis.** X-ray diffraction data were collected from frozen LCP crystals at the X06SA (PXI) beamline at the Swiss Light Source (SLS) of the Paul Scherrer Institute (PSI, Villigen, Switzerland) on an EIGER 16M detector. Promising crystals were identified using a grid-scan protocol implemented in the SSX suite[72]. From these crystals, 109 partial datasets (minisets) were recorded, which were then indexed and integrated with

*XDS*[73]. Twenty-seven minisets, ranging between 15° and 32° wedge angle, were selected for the similarity in their cell parameters (to exclude possibly occurring non-isomorphous crystals) and were scaled with *AIMLESS*[74] within the CCP4 package[75]. As a deltaB value of 17 Å$^2$ in *AIMLESS*[74] indicated moderate anisotropy, the scaled data were anisotropy-corrected and merged using the STAR-ANISO server[46] from Global Phasing Ltd. The structure of $\alpha_{1B}AR_{XTAL}$ was determined by molecular replacement with the help of *PHASER*[76] using the coordinates of turkey $\beta_1AR$ (PDB ID: 6IBL) and DARPin 5m3_D12 (PDB ID: 5LW2) as the search model, respectively. The single solution from *PHASER*[76] was refined by multiple rounds of model building in *COOT*[77] and maximum likelihood refinement with *BUSTER*[78] and *REFMAC*[79]. Validation during the course of refinement was performed using MolProbity[80]. Statistics for data collection and refinement can be found in Supplementary Table 3. A summary of the geometrical quality of the model is reported here below.

The overall MolProbity[80] score is 1.76 and the all-atom clashscore is 5. In addition, 94.2% of the residues are within Ramachandran favored regions and 5.6% within allowed regions. There is only one Ramachandran plot outlier (D357) in the unstructured linker between GPCR and DARPin crystallization chaperone. Finally, there are four side-chain rotamer outliers (F202, Y223, F284, and Q524, i.e., only ~1% of the analyzed side chains). There are no such rotamer outliers in the ligand-binding site; one outlier is in the crystallization chaperone (Q524), while two of the remaining three outliers are immediately preceding or following an unstructured region (F202 and F284).

Receptors were structurally aligned using the command "align" in PyMOL (version 2.4.0a0), allowing five cycles of refinement on all atoms or on backbone atoms unless otherwise stated. For the $\alpha_{2C}AR$-RS79948 structure (PDB ID: 6KUW[26]), the coordinates of chain A were used for structural analysis. The OBS in the $\beta_2AR$-epinephrine complex consists of residues 3.32, 3.33, 3.36, 5.42, 5.46, 6.48, 6.51, 6.52, 6.55, 7.39, and 7.43[16].

**Mammalian cell culture**. HEK293T/17 cells (American Type Culture Collection) were maintained in Dulbecco's modified medium (Sigma) supplemented with 100 units/ml penicillin, 100 µg/ml streptomycin (Sigma), and 10% (v/v) fetal calf serum (BioConcept). Cells were maintained at 37 °C in a humidified atmosphere of 5% $CO_2$, 95% air.

**Ligand-binding assays**. Ligand-binding experiments were performed on transiently transfected HEK293T/17 cells using a homogeneous time-resolved fluorescence (HTRF) binding assay. All receptor variants were cloned into a mammalian expression vector containing an N-terminal SNAP tag (Cisbio). The construct $\alpha_{1B}AR_{XTAL}$ used for ligand-binding experiments is otherwise identical to the one used for crystallization (see above), including a cleaved 3C protease site (GP) followed by a short linker (GS) at the N terminus, and a short linker (TRE) followed by a cleaved 3C protease site (LEVLFQ) at the C terminus. This design also applies to the construct $\alpha_{1B}AR_{XTAL}$-$\Delta$D12. However, in this construct, DARPin D12 was not fused to TM7 of the receptor, and the receptor was instead truncated at G369 after helix 8. Wild-type $\alpha_{1B}AR$ and the chimeric $\alpha_{1B}AR$-$\alpha_{2C}$ mutants were N-terminally truncated to start at residue S35 to provide spatial proximity between the SNAP tag and the fluorescent ligand used as the tracer. Analogously, wild-type $\alpha_{2C}AR$ was N-terminally truncated to start at residue A37. The wild-type receptors and the chimeric $\alpha_{1B}AR$-$\alpha_{2C}$ mutants harbored the wild-type full-length C terminus.

For the HTRF ligand-binding assay, the receptors were labeled by covalently linking a Lumi4-Tb fluorophore (Cisbio) to the N-terminal SNAP tag of the receptor. Lumi4-Tb was used as a FRET (Förster resonance energy transfer)-donor (excitation at 340 nm and emission at 490 nm and 620 nm). The $\alpha_1AR$ ligand QAPB was used as a FRET-acceptor (excitation at around 490 nm and emission at 520 nm). If donor and acceptor are in spatial proximity and the donor is excited at 340 nm, the donor's emission at 490 nm is quenched by the acceptor, whose emission at 520 nm is increased. This setup enabled the simultaneous measurement of receptor bound to the ligand QAPB (emission intensity at 520 nm) and total receptor (emission intensity at 620 nm). Both together allowed us to normalize binding to the receptor expression levels as well as to determine the relative receptor expression levels in our cell samples.

HEK293T/17 cells were harvested by trypsinization. Cells were reverse-transfected with a mix of 863 ng/ml DNA and 15.43 µl/ml TransIT–293T® (Mirus Bio) in Opti-MEM (Gibco), which was incubated for 20–30 min at room temperature and subsequently 471.6 µl thereof supplemented to 6 ml of a cell resuspension containing ~185,000 cells/ml. This cell mix was dispensed to poly-D-lysine (Gibco)-coated 384-well plates (Greiner 781080) with 40 µl/well and incubated 46–50 h at 37 °C and 5% $CO_2$. Medium was discarded, 15 µl/well 50 nM SNAP-Lumi4-Tb (Cisbio) in assay buffer (20 mM HEPES pH 7.5, 100 mM NaCl, 3 mM $MgCl_2$, 0.05% (w/v) BSA, 0.2% (w/v) skim milk) were added, and the plate was incubated at 37 °C for 1.5 h. The solution was discarded, and the cells were washed four times with 50 µl/well ice-cold assay buffer. For saturation ligand-binding experiments, labeled cells were incubated with a dilution series of QAPB (ThermoFisher Scientific) in assay buffer ranging from $1 \times 10^{-6}$ M to $2.92 \times 10^{-11}$ M. For competition ligand-binding experiments, labeled cells were incubated with a serial dilution of cyclazosin (Sigma-Aldrich), RS79948 (Tocris), or prazosin (Tocris) prepared in assay buffer, in the presence of 2 nM QAPB for

constructs $\alpha_{1B}AR$, $\alpha_{1B}AR$-$\alpha_{2C}$(Y3.28), $\alpha_{1B}AR$-$\alpha_{2C}$(L45.52), $\alpha_{1B}AR$-$\alpha_{2C}$(Y6.55), $\alpha_{1B}AR$-$\alpha_{2C}$(YLLY;N2.64;G7.32), $\alpha_{1B}AR$-$\alpha_{2C}$(YLLY;N2.64;C5.43), $\alpha_{1B}AR_{XTAL}$, and $\alpha_{1B}AR_{XTAL}$-$\Delta$D12, or 50 nM QAPB for $\alpha_{2C}AR$, $\alpha_{1B}AR$-$\alpha_{2C}$(L3.29), $\alpha_{1B}AR$-$\alpha_{2C}$(YLLY), $\alpha_{1B}AR$-$\alpha_{2C}$(YLLY;ECL2), and $\alpha_{1B}AR$-$\alpha_{2C}$(YLLY;N2.64). After incubation on ice for 2–4 h, emission intensities at 520 nm and 620 nm after an excitation at 340 nm were measured on a SPARK fluorescence plate reader (Tecan). The ratio of emission intensities of FRET-acceptor and FRET-donor (Em520 nm/Em620 nm) was calculated, and $K_D$ and $IC_{50}$ values were obtained by fitting the data with a four-parameter non-linear regression with *GraphPad Prism* Suite 8.4.3. $K_i$ values were calculated using the Cheng-Prusoff equation[81].

**Signaling assays**. Signaling assays were performed with receptor constructs harboring the wild-type helix 8 and the full-length C-terminus as well as the entire ICL3. Agonist-induced IP1 accumulation was measured in transiently transfected HEK293T/17 cells, as described before[82]. Twenty-four hours after transfection, cells were washed with phosphate-buffered saline (PBS), detached with trypsin-EDTA (Sigma), and resuspended in assay buffer (10 mM HEPES pH 7.4, 1 mM $CaCl_2$, 0.5 mM $MgCl_2$, 4.2 mM KCl, 146 mM NaCl, 50 mM LiCl, 5.5 mM glucose, 0.1% (w/v) BSA). Cells were seeded at 20,000 cells per well in white 384-well plates (Greiner) and incubated for 2 h at 37 °C with a concentration range of phenylephrine (Tocris) diluted in assay buffer. IP1 accumulation was measured using the HTRF IP-One kit (Cisbio) according to the manufacturer's protocol on a SPARK fluorescence plate reader (Tecan). This kit uses an anti-IP1 antibody labeled with Tb-cryptate as FRET-donor (excitation at 317 nm, emission at 620 nm) and IP1-d2 as FRET-acceptor (emission at 655 nm). The ratio of emission intensities of FRET-acceptor and FRET-donor (Em655 nm/Em620 nm) was calculated, and $EC_{50}$ (median effective concentration) values were obtained by fitting the data with a three-parameter non-linear regression with *GraphPad Prism* Suite 8.4.3.

**Docking and molecular dynamics (MD) simulations**. For MD simulations of $\alpha_{1B}AR$, the crystallographic pose of (+)-cyclazosin shown in Fig. 2a on the left was used as the starting pose. For $\alpha_{2C}AR$ (PDB ID: 6KUW[26]), the initial poses of the non-crystallographic ligands yohimbine and corynanthine were generated using the docking function of ICM-Pro (Molsoft LLC). The standard docking procedure involved defining receptor atoms within 5 Å of the co-crystallized ligand RS79948 as the binding site of interest, around which a 25 × 25 × 25 Å box was established. Docking was performed while maintaining the receptor as rigid, except for corynanthine docking, where L128$^{3.29}$ needed to be flexible in order to allow corynanthine to dock into the pocket. Typically, only the most energetically favorable docking poses were analyzed with MD, but stacks of up to 80 possible poses were visually inspected as part of the docking optimization process. MD preparation and simulations were conducted using the Desmond MD simulation system[83], as previously described[84]. For MD simulations of $\alpha_{1B}AR$, the DARPin D12 fusion was excised. For MD simulations of $\alpha_{2C}AR$, the *P. abysii* glycogen synthase fusion was removed. Any loop residues that were missing from the starting structures were left out in the MD simulations. Imported protein-ligand docked structures were pre-processed and minimized with OPLS3e force fields[85] using the protein preparation wizard tool. Using the system builder tool, a 1-palmitoyl-2-oleoyl-*sn*-glycero-3-phosphocholine (POPC) bilayer was built around the receptor with an additional 10 × 10 × 10 Å orthorhombic buffer. The simple point charge (SPC) water model was used. The $\alpha_{1B}AR$ structure was neutralized by 12 Cl$^-$ ions and the $\alpha_{2C}AR$ structure by 13 Cl$^-$ ions, and Na$^+$ and Cl$^-$ ions corresponding to 0.15 M NaCl were added to the system. Protein structures were first relaxed through a series of MD simulations. Initially, Brownian Dynamics was run at 10 K with restraints on solute heavy atoms for 100 ps under NVT conditions. This was followed by two simulations at 50 K with the protein and the Z-plane of the membrane restrained, first under NPT then NPγT conditions. The next simulation involved gradual heating from 100 to 300 K with a gradual release of restraints, followed by a final relaxation step where restraints were removed. Production NPγT MD simulations were then run at 300.0 K and 1 atm for 300 ns with a recording interval of 300 ps. The membrane surface tension was set to 4000 bar/Å. MD data were gathered using VMD 1.9.3[86]. Protein and ligand root-mean-square deviation (RMSD) values were obtained using the RMSD visualizer tool. RMSD of ligands and protein backbone residues are relative to the receptor in its initial frame after the MD relaxation protocol described above.

**Reporting summary**. Further information on research design is available in the Nature Research Reporting Summary linked to this article.

## Data availability

Coordinates and structure factors for the complex of $\alpha_{1B}AR_{XTAL}$ and (+)-cyclazosin have been deposited in the worldwide PDB under the accession code: 7B6W. All data needed to evaluate the conclusions of the paper are present in the main manuscript and/or in the Supplementary Information. Additional data supporting the findings of this paper are available from the corresponding authors upon reasonable request. Additional publicly available PDB entries mentioned in this paper: 6KUW; 6KUX; 6KUY; 6K41; 2YCW; 2RH1; 4LDO; 6IBL; 5LW2. Source data are provided with this paper.

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

## Acknowledgements
We thank B. Blattmann and C. Müller-Simmen of the Protein Crystallization Center at the University of Zurich for their support during crystallization, and the staff of the X06SA beamline at the Paul Scherrer Institute for support during X-ray data collection. We thank G. Liechti-Tonarque and D. Barret for their help with protein expression, purification, and characterization, and C. Manatschal for support during the initial phase of refinement of the crystal structure. We thank M. Hilge for data processing and refinement of the crystal structure, and G. Murshudov for his help in the creation of the cyclazosin dictionary. We also would like to acknowledge V. Jameson of the Melbourne Cytometry Platform for assistance with FACS. This work was supported by Schweizerische Nationalfonds grant 31003A_182334 (to A.P.) and Australian National Health and Medical Research Council grants 1081801 and 1137179 (to D.J.S.).

## Author contributions
D.J.S. and R.R.C. carried out directed evolution on α₁BAR and characterized single clones. M.D. designed the α₁BAR-DARPin D12 fusion and the crystallization construct α₁BAR_XTAL. M.D., L. Morstein., and L. Merklinger cloned, expressed, purified, crystallized α₁BAR_XTAL, and harvested crystals. M.D. and M.S. established expression and purification protocols with support from O.Z., S.V., and P. E. M.D., L. Morstein, and A.K. collected X-ray diffraction data. M.D. supervised S.V., L. Morstein, L. Merklinger, and A.K. in the aforementioned tasks, organized and planned the experiments. P.R.E.M. contributed to X-ray data processing and refinement in the initial phase of this project. M.D. analyzed and interpreted structural data. L. Morstein cloned receptor mutants, performed ligand-binding and signaling experiments, and analyzed the data with support from C.K. and M.D. S.A.E. supported L. Morstein in the initial phase of this project. D.J.S., L.A.d.Z., D.K.C., and T.M.V. performed docking and molecular dynamics simulations. Project management was carried out by M.D. and A.P. with support from D.J.S. and O.Z. The manuscript was prepared by M.D., L. Morstein, D.J.S., and A.P. with inputs from all authors. All authors contributed to the final editing and approval of the manuscript.

## Competing interests
The authors declare no competing interests.
