## [Peer Review File · Nature Communications]

Crystal structure of the α 1B-adrenergic receptor reveals molecular determinants of selective ligand recognitionREVIEWER COMMENTS

Reviewer #1 (Remarks to the Author):

The authors present the first structure of an $\alpha 1$ adrenergic receptor, a previously missing piece for understanding adrenergic receptor ligand subtype selectivity with implications for drug (side-)effects.

In general, the paper is well written and presents the authors' conclusions in a clear and coherent manner. The structure determination itself was enabled by a combination of: mutagenesis via a directed evolution protocol developed in the authors' laboratory; a number of receptor truncations; as well as the first application of their DARPin crystallization chaperone for determination of a novel GPCR structure, a welcome technological advance. Material for structural studies was expressed in E. coli.

While the number of receptor modifications makes it apparent that this is not a 'straightforward' target for structure determination, it is somewhat unclear if perhaps a more wt-like system with fewer modifications would have been tractable if material for structural studies had been produced in eukaryotic expression systems. Was this ever attempted?

In balance, this is perhaps the most extensively engineered GPCR construct this reviewer has encountered to date and as such, it is paramount to characterize the individual modifications the authors have introduced much more carefully.

In particular, the authors should -

1. Provide functional, and not only ligand-binding, data, at least of wt; a construct with all 12 stabilizing mutations, but ideally, also constructs with single point mutant resolution; a construct with truncated ICL3; a construct without H8; and a construct with H8 substituted by their DARPin. This will complement the ligand binding data they present in the manuscript.

2. Is it correct that the xtal- Δ D12 construct used in Fig S3 and elsewhere contains H8? If so, the authors should provide an additional ligand binding data series where H8 is deleted.

3. The way their ligand binding data is presented, perhaps generously, suggests that there is only little difference (in pKi) between wt and their xtal construct. This is possibly due to the wt construct assayed also being in G protein decoupled state; the functional differences between these constructs is likely higher. The authors should therefore also provide affinity data for wt in presence and absence of G protein coupling.

4. If possible, the authors should assay ligand binding of E. coli-produced material, or alternatively, discuss possible artifacts (structure, binding, activation, missing QC machinery, etc) introduced by this expression system, in general as well as in context of their modifications (e.g. N-terminal truncation of possible glycosylation motifs etc).

In addition to acquiring these important additional data, the authors should

5. Comment on how they identified the four point mutations within their DARPin (L3K, G4A, K5R, and A13K)

6. The ligand RMSD shown in Fig 2 is on the high side (up to 2 Å RMSD over the course of their simulation) - does the ligand shift or distort, or is this perhaps, in part, due to flips of the furan moiety they mention?

7. In Fig S5, the authors should also show ligand omit maps.

8. Fig S8 indicates mobility of TM1, which is observed in artificial conformation in their structure due to crystal packing. Over the course of their simulation, does this helix eventually assume a stable orientation with respect to the helical bundle? They truncate their MD after 300 ns although it appears there is some increase in RMSD (light blue trace in panel A) after approx. 250 ns - perhaps a longer overall simulation time should be chosen?

9. Stereochemical quality of the model (PDB report) could and should be improved, based on their PDB validation report.

10. Likewise, Rmerge and Rpim seem high, and completeness low. The CC value at the highest resolution shell might indicate an optimistic choice of resolution cutoff? Overall, the stats presented in Table 1 are not good, and Rfree is too high for the claimed resolution. Are the authors able to improve them, e.g. by processing only a subset of the crystals, or more careful data processing?

11. At some point in their supplement or materials, can the authors please provide the full amino acid sequence of their DARPin? This would prevent the reader from having to cross-reference multiple papers and/or extracting these residues from the Protein Data Bank.

Reviewer #2 (Remarks to the Author):

The article by Deluigi et al., describes the crystal structure of inverse agonist-bound human α 1BAR stabilized by a DARPin. Authors compare the current structure with previously structures of α 2ARs, and complement the structural findings with biochemical experiments to decipher the potential determinants of ligand selectivity. Taken together with the previous structures of α 2ARs, the current structure now also provides a structural framework to design sub-type specific ligands. Overall, the findings are very interesting and of broad interest, experiments are well designed, data are appropriately analyzed, and the manuscript is well written. I have only a couple of minor suggestions for the authors to consider during revision.

1. The ligand binding experiments are carried out using HTRF assay on whole cells instead of conventional radioligand binding on membranes. It would be useful to mention for the readers that the binding profiles of different ligands in the HTRF assays are comparable to those previously reported in the literature using radioligand-based assays.
2. Along the same line, authors should also mention if the surface expression of different mutants are comparable and normalized in the ligand binding assays. This is an important consideration for interpreting the binding experiments carried out on intact cells.
3. Statistical analysis of the binding data is included in Table S6 at the end of the manuscript. It may be useful to mention this in the main figure legends in order to interpret the differences in the binding affinities of the ligands for different mutants.
4. The observation on L334 is intriguing and worth looking into a little further. While there is a slight loss of thermostability on back mutation to F, how does it affect cyclazosin binding? What about the binding of other ligands?

Arun K. Shukla, Ph.D.

Reviewer #3 (Remarks to the Author):

Deluigi and coworkers present the first crystal structure of the alpha1B-adrenergic receptor (alpha1B-AR) in complex with an inverse agonist. Such structural information is essential to develop selective ligands targeting this GPCR, whose physiological and pathological roles are only partially characterized.

The authors do a thorough comparison of the protein-ligand interactions observed for alpha1B-AR and other adrenergic receptor structures and propose residues 3.29 and 6.55 (Ballesteros-Weinstein numbering) as key determinants of ligand selectivity. Such prediction is further tested with site-directed mutagenesis data and ligand binding assays. The conclusions of the authors are well supported by the data and the study may be used as stepping stone to design selective ligands targeting alpha1B-AR. Therefore, I believe it merits publication, provided that a few minor issues are addressed.

- On page 7, the authors characterize the alpha1B-AR structure as apparently captured in the inactive state. Besides the comparison of the TM6 position with other adrenergic receptor structures, I would suggest that the authors attempt to quantify the % degree active (as done in GPCRdb) or the activation index (for class A GPCRs, <https://ccda250.chemie.uni-erlangen.de/a100/a100-python/public/>).

- On page 8, the authors describe a salt bridge potentially involved in activation. Do residues 7.36 and 2.65 correspond to one of the known GPCR activation microswitches? (see e.g. <https://europepmc.org/article/ppr/ppr305890>) What are the equivalent residues in other adrenergic receptors (ARs) and, in case both active and inactive structures exist for those ARs, how do the interactions between these two residues change between the two states? Have mutations at these positions (either site-directed or natural variants) reported for ARs or other aminergic receptors, which also affect receptor activation?

- On page 9, the authors explain that cyclazosin is likely to be protonated. If that is the case, I would suggest showing a protonated N1 in the 2D chemical representations of the ligand shown across the manuscript. Moreover, I understand from reference 49 that the calculated pKa of this molecule is 9. It might be worth mentioning this computational estimation to further support the consideration of a protonated molecule.

- The authors describe how cyclazosin occupies the orthosteric binding site as well as secondary binding pockets. Do any of these secondary binding pockets correspond to the allosteric/ vestibular binding site near ECL2 described for other class A GPCRs? In other words, can cyclazosin be considered a bitopic ligand that can "fill" simultaneously both the orthosteric and the allosteric binding sites?

- Do the authors observed any electron density that could correspond to one or more cholesterol molecules bound to alpha1-AR? (since cholesterol seems to be an allosteric modulator of other GPCRs, as nicely shown by some of the authors of this manuscript for the oxytocin receptor: <https://doi.org/10.1126/sciadv.abb5419>)

- The authors should provide more details regarding their docking protocol: was the receptor considered as rigid or flexible? (this could affect the clashes described in the manuscript); did they include

information about the binding site in their docking protocol, in terms of initial position of the crystallographic ligand and/or putative binding residues?; how many docking poses were generated? was only the top scored pose analyzed or all of them?

- Regarding the molecular dynamics (MD) details, the authors ran simulations for both the crystallographic construct and a structure reverting the L334(7.39) to F mutation. Did they also revert the other thermostabilizing mutations introduced in the construct to facilitate crystallization (in particular the partial replacement of TM7 by the linker)? And how were the missing residues in the ICLs modeled?

- I find commendable that the authors decided to run MD simulations to assess the impact of the F334(7.39) to L mutation. Considering their effort and the length of their simulations (~300 ns), I would suggest that they try to exploit this computational data for further analysis. Does the mutation affect the protein flexibility, in terms of RMSF? F334(7.39) is located one helical turn away from Y338(7.43), which in turns interact with the essential D125(3.32). Can the authors check whether there is some stacking interaction between F334 and Y338 and if the H-bond between Y338 and D125 (shown in Figure 2a) is maintained after reverting the mutation? I would also be curious to know more about the behavior of V197(45.52), whose side chain was only partially resolved in the X-ray structure.

- I also liked very much that the authors honestly describe the possible limitations of the different modifications introduced in the crystallographic construct when describing the structure. The only thing not fully clear to me is how the linker inserted in place of the intracellular part of TM7 could affect the position of the adjacent TM6 (which in turn the authors used to describe the activation state of the receptor).

- Although probably beyond the scope of this manuscript, have the authors considered to use their MD trajectories to estimate the difference in ligand binding free energy between the XTAL and MD receptors? And use per-residue decomposition in order to e.g. see if residues 3.29 and 6.55 contribute not only to selectivity, but also significantly to binding free energy?

Responses to reviewers' comments are in blue.

Line numbering refers to the revised version of the manuscript.

REVIEWER COMMENTS

Reviewer #1 (Remarks to the Author):

The authors present the first structure of an $\alpha 1$ adrenergic receptor, a previously missing piece for understanding adrenergic receptor ligand subtype selectivity with implications for drug (side-)effects. In general, the paper is well written and presents the authors' conclusions in a clear and coherent manner. The structure determination itself was enabled by a combination of: mutagenesis via a directed evolution protocol developed in the authors' laboratory; a number of receptor truncations; as well as the first application of their DARPin crystallization chaperone for determination of a novel GPCR structure, a welcome technological advance. Material for structural studies was expressed in *E. coli*.

We thank the reviewer for the appreciation of our manuscript and the DARPin crystallization chaperone.

While the number of receptor modifications makes it apparent that this is not a 'straightforward' target for structure determination, it is somewhat unclear if perhaps a more wt-like system with fewer modifications would have been tractable if material for structural studies had been produced in eukaryotic expression systems. Was this ever attempted?

Expression in a eukaryotic system was not attempted, as constructs were available that were well expressed in *E. coli* as properly folded, ligand-binding competent receptors with no hints of altered properties compared to expression in eukaryotic cells (see response to point 4 below). We recognize the value of eukaryotic systems for heterologous overexpression of GPCRs for structural studies. However, the receptor modifications we introduced are essential to improve *crystallizability*, and thus, they are required regardless of the expression host. Such modifications (truncation of long flexible regions; fusion of a crystallization chaperone like the DARPin used here; stabilizing mutations) are present in most GPCR constructs whose structures have been crystallographically determined to date, regardless of bacterial or eukaryotic expression. Determination of GPCR crystal structures without such modifications proved to be nearly

impossible, with only limited examples, usually receptors with particularly favorable properties (e.g., high intrinsic stability, short ICL3 or termini), or where conformational stabilization and crystal contacts were provided by a binding protein (e.g., antibodies or nanobodies). We now provide a short explanation in lines 372–374.

A more detailed discussion of the stabilizing mutations is presented below.

In balance, this is perhaps the most extensively engineered GPCR construct this reviewer has encountered to date and as such, it is paramount to characterize the individual modifications the authors have introduced much more carefully. In particular, the authors should –

1. Provide functional, and not only ligand-binding, data, at least of wt; a construct with all 12 stabilizing mutations, but ideally, also constructs with single point mutant resolution; a construct with truncated ICL3; a construct without H8; and a construct with H8 substituted by their DARPin. This will complement the ligand binding data they present in the manuscript.

We apologize for the initial lack of additional functional data, which we have now included in the revised manuscript.

We observed that G protein signaling of the stabilized mutant is impaired compared to the wt receptor. The stabilized mutant is thus locked in an inactive state, which we now point out in the Results in lines 110–112 and in the Discussion in line 375, along with a description of the signaling assays in the Methods in lines 634–649. This result is not unexpected, as a reduction or loss of signaling competency is a common outcome of GPCR stabilization for crystallography (e.g.: Warne, T. *et al.*, *Nature* **454**, 486-491 (2008); White, J. F. *et al.*, *Nature* **490**, 508-513 (2012); Tate, C. G., *Trends Biochem. Sci.* **37**, 343-352 (2012)). It is even more likely to occur when the stabilization procedure is carried out in the presence of an antagonist or inverse agonist, which keeps the receptor in an inactive conformation, as is the case in our study.

We agree that the interpretation of conformational changes required for receptor activation is limited from inactive state structures. Nonetheless, it is widely recognized that the structures of such stabilized receptors provide extremely valuable insights into the ligand binding modes of the co-crystallized ligands, as well as insights into how ligand binding affects side-chain conformations within the binding pocket.

In the present study, we describe the binding mode of an inverse agonist in an inactive-state $\alpha_{1B}AR$ structure. The inactive-state structure is thus fully consistent with the pharmacological properties of the ligand. Crucially, the crystallized construct retained near-native affinity for the co-crystallized ligand. Overall, the entire focus of this study are inactive-state adrenergic receptor structures bound to *antagonists* or *inverse agonists*. The $\alpha_{1B}AR$ structure presented here enabled us to enrich the knowledge of ligand binding modes and selectivity determinants within the ligand-binding site, which did not require the formation of a functional complex with signaling proteins.

As suggested by the reviewer, we have additionally measured the impact of each of the 12 mutations on signaling competency (depicted below). We will present these data in an upcoming manuscript that describes the design of a signaling-competent mutant. Such a mutant, albeit substantially less stable and thus not suitable for crystallography, may provide insights into receptor activation by means of other biophysical methods. Receptor activation is, however, beyond the scope of the present manuscript for the reasons explained above.

We found that the following single mutations substantially impair agonist-induced G protein signaling: S150Y, G183V, T295M, V333L, F334L, and P349L. However, as this is not the focus of the present manuscript, we do not wish to discuss these mutations here.

	EC ₅₀ (log M)		E _{max} (% of wt)	
	mean	SEM	mean	SEM
wt	-6.45	0.11	100.0	
S95C	-5.67	0.11	85.6	4.7
I116T	-6.35	0.19	85.2	7.3
V124M	-6.57	0.16	79.5	6.0
S150Y	-5.31	0.25	14.6	2.0
S168C	-6.30	0.15	127.9	8.8
G183V	-5.11	0.91	3.9	2.0
D191Y	-5.68	0.09	102.4	4.3
E194V	-6.10	0.27	86.2	10.8
T295M	-5.45	0.37	36.1	6.9
V333L	-5.47	0.15	72.6	5.5
F334L	n.a.		n.a.	
P349L	-5.45	0.59	17.0	5.1

Impact of each of the 12 mutations in $\alpha_{1B}AR_{XTAL}$ on agonist-induced G_q signaling. The upper graphs represent dose-response curves for wild-type (wt) $\alpha_{1B}AR$ as well as for $\alpha_{1B}AR$ constructs harboring the individual mutation indicated. Phenylephrine is a full agonist at $\alpha_{1B}AR$. The table summarizes the corresponding EC₅₀ and E_{max} values. *n.a.*, non-applicable because the upper plateau could not be reached. A substantial impairment of signaling was defined as a drop in E_{max} compared to wt, together with a drop in agonist potency of $\geq \sim 10$ -fold.

In most cases, stabilizing mutations are required to conformationally stabilize (i.e., “rigidify”) the receptor. The high conformational flexibility of most wt GPCRs is crucial for signaling; however, it is also a recognized major obstacle for crystallographic studies (e.g.: Tate, C. G., *Trends Biochem. Sci.* **37**, 343-352 (2012); Heydenreich, F. M. *et al.*, *Front. Pharmacol.* **6**, 82 (2015); Tehan, B. G. & Christopher, J. A., *Curr. Opin. Pharmacol.* **30**, 8-13 (2016)).

Importantly, we have previously indeed reverted some of the above-mentioned mutations and attempted to obtain crystals. However, these attempts remained unsuccessful, and we observed a correlation between the loss of thermostability (Supplementary Fig. 3e and curves depicted below) — and thus likely of conformational rigidity — and the inability to crystallize. The correlation between thermostability and GPCR crystallizability has been well documented before (e.g.: Tate, C. G., *Trends Biochem. Sci.* **37**, 343-352 (2012); Heydenreich, F. M. *et al.*, *Front. Pharmacol.* **6**, 82 (2015); Tehan, B. G. & Christopher, J. A., *Curr. Opin. Pharmacol.* **30**, 8-13 (2016)).

In addition, we extensively tried to crystallize, albeit unsuccessfully, a construct with only six mutations (derived from the α_{1B} AR mutant described by Yong, K. J. *et al.*, *ACS Chem. Biol.* **13**, 1090-1102 (2018)). This construct lacks the signaling-impairing mutations G183V, V333L, and P349L, but appeared to be too unstable and likely conformationally too flexible to yield crystals.

Loss of thermostability upon reversion of the V333L mutation to the wt valine residue. CPM-based thermostability assay of the α_{1B} AR-B1 mutant selected by directed evolution, harboring the V333L mutation (solid blue circles), and of α_{1B} AR-B1-V333, which includes the reversion of V333L to valine (open orange diamonds). Both constructs were bound to the inverse agonist prazosin. The apparent melting temperature (T_m) was estimated from these data using non-linear regression with GraphPad Prism 8.4.3. Data from a representative experiment are shown.

Finally, we agree that the ICL3 deletion and the replacement of H8 by the DARPin fusion are likely to interfere with binding of a G protein. These modifications should be thus avoided in studies involving signaling proteins (we of course carried out our signaling studies in the absence of such modifications). However, to reiterate, the focus of this study is an inactive-state receptor decoupled from the G protein. We have now clarified this point in the Discussion in lines 381–383. Thus, we believe that additional functional experiments involving the above-mentioned modifications would not be required considering the scope of this study.

2. Is it correct that the xtal- Δ D12 construct used in Fig S3 and elsewhere contains H8? If so, the authors should provide an additional ligand binding data series where H8 is deleted.

In Supplementary Fig. 3c and d, we present binding data of the co-crystallized ligand, cyclazosin, to three constructs:

- wt $\alpha_{1B}AR$, i.e., with H8.

- $\alpha_{1B}AR_{XTAL}$: engineered construct without H8 (H8 is deleted and replaced by DARPin D12).

- $\alpha_{1B}AR_{XTAL}-\Delta D12$: engineered construct with H8 (i.e., without DARPin D12).

Our data thus already indicate that the deletion of H8 does not affect cyclazosin affinity, suggesting that the omission or replacement of H8 by the DARPin does not perturb the antagonist- / inverse agonist-binding pocket, which is the focus of the present study.

Analogously, we have previously reported that the same fusion of DARPin D12 to the neurotensin receptor 1 neither perturbed affinities to agonists or inverse agonists, nor was there any relevant structural deviation in the seven-transmembrane bundle, compared to the same receptor mutant crystallized without DARPin (Deluigi, M. *et al.*, *Sci. Adv.* **7**, eabe5504 (2021)).

3. The way their ligand binding data is presented, perhaps generously, suggests that there is only little difference (in pKi) between wt and their xtal construct. This is possibly due to the wt construct assayed also being in G protein decoupled state; the functional differences between these constructs is likely higher. The authors should therefore also provide affinity data for wt in presence and absence of G protein coupling.

The reviewer refers to the ligand-binding data we acquired for the co-crystallized ligand, cyclazosin. Cyclazosin is a potent *inverse* agonist. As a consequence, it is expected to stabilize $\alpha_{1B}AR$ in the inactive, G protein-decoupled state. Therefore, we can only provide affinity data for wt $\alpha_{1B}AR$ bound to cyclazosin in a G protein-decoupled state.

Our ligand-binding data demonstrate that the crystallized construct binds cyclazosin with a very similar affinity as wt $\alpha_{1B}AR$ (see lines 117–119 and Supplementary Fig. 3d). The slight difference in affinity compared to wt is within the range of what has been observed for many crystallized GPCR constructs (for instance.: Chien, E. Y. T. *et al.*, *Science* **330**, 1091-1095 (2010); Wang, C. *et al.*, *Science* **340**, 610-614 (2013); Wang, S. *et al.*, *Science* **358**, 381-386 (2017); Schöppe, J. *et al.*, *Nat. Commun.* **10**, 17 (2019); Shiimura, Y. *et al.*, *Nat. Commun.* **11**, 4160 (2020)). Consequently, it appears that receptor engineering did not substantially perturb cyclazosin binding. This is consistent with the observation that mutagenesis locked the receptor in an inactive state (see above). We believe that our ligand-binding data, together with the MD simulations, where we reverted some of the

mutations, provide a solid indication that receptor engineering did not affect the conclusions of this study.

If the reviewer was referring to affinity data for binding of agonists, please see below comment 4 of reviewer #2. However, agonists are not the focus of the present study for the reasons explained above.

4. If possible, the authors should assay ligand binding of *E. coli*-produced material, or alternatively, discuss possible artifacts (structure, binding, activation, missing QC machinery, etc) introduced by this expression system, in general as well as in context of their modifications (e.g. N-terminal truncation of possible glycosylation motifs etc).

(i) There is experimental evidence that ligand binding is not different for $\alpha_{1B}AR$ expressed in *E. coli* or in mammalian cells. We have now mentioned this reassuring consideration in lines 378–379.

We have previously reported the affinity of prazosin — a ligand nearly identical to the co-crystallized cyclazosin — for $\alpha_{1B}AR$ expressed in *E. coli* (Schuster, M. *et al.*, *Biochim. Biophys. Acta Biomembr.* **1862**, 183354 (2020)). We determined a sub-nM affinity (0.074 nM), which agrees very well with the affinity range reported in the literature for $\alpha_{1B}AR$ expressed in mammalian cells (0.05–0.55 nM, with a mean of 0.11 nM as reported by Docherty, J. R., *Eur. J. Pharmacol.* **855**, 305-320 (2019)).

(ii) We have recently shown that the ligand-binding site of the neurotensin receptor 1 (NTSR1) is virtually identical in structures obtained from receptors produced in *E. coli* or *Sf9* insect cells (see picture below) (Deluigi, M. *et al.*, *Sci. Adv.* **7**, eabe5504 (2021)).

Comparison of the ligand-binding site of NTSR1 produced in *E. coli* (yellow) and *Sf9* insect cells (slate) reveals no differences. Panel A shows that the binding site is virtually identical. $RMSD_{C\alpha} \approx$ only 0.7 Å, including ECL1–3. The side chains adopt highly similar conformations in both structures (V234L is a mutation present in the receptor expressed in *E. coli*). Panel B shows that the binding mode of the peptide agonist NTS₈₋₁₃ is also virtually identical in both structures.

(iii) Despite the lack of a eukaryotic QC machinery, $\alpha_{1B}AR$ could be successfully expressed in the inner membrane of *E. coli* as a properly folded, ligand-binding receptor. Furthermore, we used a prazosin ligand-affinity column to purify $\alpha_{1B}AR$ expressed in *E. coli*. We have now added a note to this regard in lines 112–114.

In general, there are many examples of successful *E. coli* expression of eukaryotic membrane proteins, including GPCRs, provided that the proper expression construct and conditions are determined. For a review about GPCRs, see e.g.: Abiko, L. A. *et al.*, *J. Biomol. NMR* **75**, 25-38 (2021). Interestingly, the study of Abiko *et al.* included a comparison of β_1AR produced either in *E. coli* or insect cells. The authors demonstrated that “the *E. coli* sample quality is identical to the insect cell material as assayed by the thermal melting behavior and a direct comparison of backbone 1H - ^{15}N spectra of the same receptor constructs obtained from insect cells in 3 different functional states. As such, the *E. coli*-derived β_1AR is usable for any structural or biophysical study in the same way as β_1AR derived from insect cells”.

In addition to acquiring these important additional data, the authors should

5. Comment on how they identified the four point mutations within their DARPIn (L3K, G4A, K5R, and A13K)

The software suite Rosetta (Leaver-Fay, A. *et al.*, *Methods Enzymol.* **487**, 545-574 (2011)) was used to optimize the sequence of the linker between GPCR and DARPin as well as the DARPin N-terminus. The aim was to design a fusion site between GPCR and DARPin featuring a compromise between rigidity and malleability in order to adapt to different crystal packings. We have now added this piece of information to the Methods section in lines 464–466.

We will describe in detail how we optimized the DARPin fusion strategy in an upcoming manuscript.

6. The ligand RMSD shown in Fig 2 is on the high side (up to 2 Å RMSD over the course of their simulation) - does the ligand shift or distort, or is this perhaps, in part, due to flips of the furan moiety they mention?

The MD simulations we carried out on both the crystal structure and the L333V-L334F back-mutant validated the crystallographically determined binding mode of (+)-cyclazosin. The ligand remained stable within the binding site throughout both simulations. Comparison of the simulations suggests that the V333L and F334L mutations do not significantly affect the binding pose of (+)-cyclazosin.

Based on the reviewer's comment #9, these simulations were repeated using the further refined crystal structure and an L333V-L334F back-mutant thereof. The results of these simulations have been inserted into the revised manuscript, replacing Fig. 2b and Supplementary Fig. 8.

(+)-Cyclazosin maintains its binding pose through the duration of the 300 ns crystal structure simulation with an average RMSD of 1.2 Å, with no noticeable shifts or distortions. Upon backmutation of V333L and F334L, the average RMSD of (+)-cyclazosin corresponds to 1.7 Å. The slightly higher RMSD is mostly due to (+)-cyclazosin undergoing a slight and transient rigid body rotation toward the extracellular regions of the receptor during the first part of the simulation. In both simulations, the

ligand equilibrates toward a nearly identical binding pose, which is very similar to the crystallographically determined one (Fig. 2b). We do not wish to speculate about the slight rigid body shift in our manuscript because (i) it is transient and possibly an artifact of the simulation, (ii) it is very small (in the same range as the coordinates' error), and (iii) because the scope of the simulations was the validation of the ligand binding mode. The transient and slight shift does not alter how the ligand is accommodated in the receptor sub-pockets, and nearly identical interactions to those reported are formed.

7. In Fig S5, the authors should also show ligand omit maps.

If the reviewer is concerned about model bias, we can reassure that the Fo–Fc map shown in Supplementary Fig. 5a (green mesh) is *before* the ligand was ever modeled in the electron density, which we have now stated in the figure caption. Consequently, there is no model bias for the ligand in the Fo–Fc map we show.

We believe that an Fo–Fc map *before* the ligand was modeled is the ideal map to be shown. However, if the reviewer would like to see an omit map for another reason, we kindly ask to specify which type of omit map we should calculate.

8. Fig S8 indicates mobility of TM1, which is observed in artificial conformation in their structure due to crystal packing. Over the course of their simulation, does this helix eventually assume a stable orientation with respect to the helical bundle?

TM1 displays flexibility in the MD simulations, without assuming a stable orientation with respect to the helical bundle. In the absence of crystal packing interactions, we expect TM1 of $\alpha_{1B}AR$ to adopt a similar position as observed in other inactive-state adrenergic receptor structures (Supplementary Fig. 7). Of note, residues belonging to TM1 do not contribute to the ligand-binding pocket, which is the focus of the present study. The scope of the simulations was to validate the crystallographically determined ligand binding mode by comparing the crystal structure with the L333V-L334F back-mutant. We believe that the position and flexibility of TM1 during the simulations do not affect our conclusions.

They truncate their MD after 300 ns although it appears there is some increase in RMSD (light blue trace in panel A) after approx. 250 ns - perhaps a longer overall simulation time should be chosen?

The flexibility of TM1 (see previous comment), together with the loops, is the reason for the moderate fluctuation of the backbone RMSD over time. When only TM2–7 are considered, the RMSD is very stable and mostly below 1.5 Å (Supplementary Fig. 8a, salmon and teal traces, and now also mentioned in line 191). Consistent with the ligand-binding site being nearly entirely constituted by residues of TM2–7, the ligand also displays high structural stability over time (Supplementary Fig. 8b). Furthermore, a trend to increasing RMSD for TM2–7 was not seen when the simulations were performed on the newly refined crystal structure. Therefore, considering the scope of these simulations (see comments 6 and 8 above), no increase in simulation time appears necessary.

9. Stereochemical quality of the model (PDB report) could and should be improved, based on their PDB validation report.

We thank the reviewer for pointing this out. We were able to significantly improve both the clashscore (from ~7 to 5) and the side chain outliers (from 2.8% to 1.3%). We have now deposited new coordinates (see new PDB validation report).

No further improvement appears to be supported by the quality of the data. Every slight improvement of one score was accompanied by worsening of another. In general, we believe that the geometrical quality of the model is now reasonably good:

- i) The overall MolProbity score is 1.76, which is good to very good.
- ii) There is only one Ramachandran plot outlier, and this outlier is in an unstructured region that belongs to the crystallization chaperone and not to the GPCR.
- iii) There are only four side chain rotamer outliers (i.e., less than 1% of all analyzed side chains), which gives a good score in comparison to all the X-ray entries deposited in the PDB and those of similar resolution (percentile scores of 69 and 88%, respectively).

There are no outliers in the ligand-binding site. One outlier is in the crystallization chaperone. Two of the remaining three outliers are immediately preceding or following an unstructured region (F202 and F284).

- iv) The clashscore is now fairly good, especially compared to the X-ray entries of similar resolution.

In addition, nearly all the RSRZ outliers are in unstructured regions or preceding or following such regions. There are no RSRZ outliers in the ligand-binding site. Five RSRZ outliers are in regions belonging to the crystallization chaperone.

10. Likewise, Rmerge and Rpim seem high, and completeness low. The CC value at the highest resolution shell might indicate an optimistic choice of resolution cutoff? Overall, the stats presented in Table 1 are not good, and Rfree is too high for the claimed resolution. Are the authors able to improve them, e.g. by processing only a subset of the crystals, or more careful data processing?

For serial crystallography data, we specifically developed a suite of in-house python scripts with which the recorded partial datasets (minisets) from several crystals initially go through two rounds of XDS processing and rigorous data quality assessment in AIMLESS. In this procedure, regions of low quality or bad data are eliminated. Next, we evaluate all possible combinations of the remaining minisets in a brute-force approach. The best isotropically merged dataset that we could obtain displays a resolution of 3.1 Å and an overall completeness of 97%. As our data are slightly anisotropic ($\Delta B \approx 16 \text{ \AA}^2$), we corrected for this anisotropy using the STARANISO server of GlobalPhasing Ltd. The resulting dataset extends to higher resolution (2.87 Å) but shows markedly lower completeness and somewhat low $CC_{1/2}$ values. Despite these less nice data statistics, for us the ultimate criterion for using these anisotropy-corrected structure factors are the substantially better maps. To inform the reader, we have now added a statement to these regards in lines 124–127.

Furthermore, we would like to refer to two statements to be found in the Frequently Asked Questions section of the STARANISO server, see https://staraniso.globalphasing.org/staraniso_FAQ.html:

Statement 1) The PDB does not compute the correct value of the completeness for anisotropic data, for the simple reason that it does not currently capture any of the information required to determine the anisotropic cut-off.

Statement 2) For anisotropic data, $CC_{1/2}$ is not suitable as a metric, so only the mean $I/\sigma(I)$ is used.

The seemingly high Rmerge value is also a direct consequence of merging several minisets, as described by Karplus, P. A. & Diederichs, K. Assessing and maximizing data quality in macromolecular crystallography. *Curr. Opin. Struct. Biol.* **34**, 60-68 (2015). According to Karplus, P. A. *et al.*, Rmerge and Rpim are not valid indicators for serial crystallography data.

Nonetheless, we reprocessed our data and tried to improve on the data reduction:

1) To the presently very conservatively selected minisets which compose our deposited dataset, we added the next best seven and twelve minisets, respectively. While completeness and resolution of these new datasets improve, Rfree values during refinement get approximately 1–1.5% worse.

2) We manually cut the anisotropy-corrected data back to 3.0 and 3.1 Å resolution, respectively. Similarly, while resolution and $CC_{1/2}$ values improve, refinement statistics get worse.

3) An attempt to only work with isotropic data yielded worse refinement statistics and poorer maps.

Taken together, all our effort to further improve on the data were unsuccessful. Nonetheless, the maps are of good quality, especially for the ligand binding site coordinating (+)-cyclazosin (see Supplementary Fig. 5), which is the focus of the present study.

11. At some point in their supplement or materials, can the authors please provide the full amino acid sequence of their DARPin? This would prevent the reader from having to cross-reference multiple papers and/or extracting these residues from the Protein Data Bank.

We thank the reviewer for this suggestion. We have now added the DARPin sequence, together with the sequence of the entire crystallized construct, to Supplementary Fig. 2c (as text, so that it can be easily copied from the final PDF document). Some additional details have been added to the Methods section to make the construct design clearer, see highlighted text in lines 467–478.

Reviewer #2 (Remarks to the Author):

The article by Deluigi et al., describes the crystal structure of inverse agonist-bound human α_1 BAR stabilized by a DARPin. Authors compare the current structure with previously structures of α_2 ARs, and complement the structural findings with biochemical experiments to decipher the potential determinants of ligand selectivity. Taken together with the previous structures of α_2 ARs, the current structure now also provides a structural framework to design sub-type specific ligands. Overall, the findings are very interesting and of broad interest, experiments are well designed, data are appropriately analyzed, and the manuscript is well written. I have only a couple of minor suggestions for the authors to consider during revision.

We thank the reviewer for the appreciation of our manuscript.

1. The ligand binding experiments are carried out using HTRF assay on whole cells instead of conventional radioligand binding on membranes. It would be useful to mention for the readers that the binding profiles of different ligands in the HTRF assays are comparable to those previously reported in the literature using radioligand-based assays.

For the ligand RS79948, we already mention in lines 244–245 that the affinities measured in the HTRF assay for wt α_1 BAR and α_2 CAR nicely agree with previous studies carried out with conventional radioligand binding assays (RLBAs).

We have now added a similar statement and the corresponding references for cyclazosin and prazosin in lines 320–321. The affinities agree very well with the literature, with maximally ~3-fold differences in affinity. Such minor differences can be expected when very different protocols are used. Some literature values are only available for rat tissue containing all three α_2 AR subtypes (noted below), which may also explain the small differences.

Here is a summary of the affinities as pK values:

- cyclazosin- α_1 BAR: 8.99 (HTRF), 8.68 (RLBA) (Proudman, R. G. W., *et al.*, *Pharmacol. Res. Perspect.* **8** (2020)).

- cyclazosin- α_2 CAR: 6.47 (HTRF), 6.17* (RLBA) (Giardinà, D. *et al.*, *J. Med. Chem.* **39**, 4602-4607 (1996)). *Measured on rat cerebral cortex tissue containing all three α_2 AR subtypes.

- prazosin- α_{1B} AR: 9.47 (HTRF), 9.26–10.3 (RLBA) (Docherty, J. R., *Eur. J. Pharmacol.* **855**, 305-320 (2019)).

- prazosin- α_{2C} AR: 7.39 (HTRF), 6.83* (RLBA) (Giardinà, D. *et al.*, *J. Med. Chem.* **39**, 4602-4607 (1996)). *Measured on rat cerebral cortex tissue containing all three α_2 AR subtypes.

2. Along the same line, authors should also mention if the surface expression of different mutants are comparable and normalized in the ligand binding assays. This is an important consideration for interpreting the binding experiments carried out on intact cells.

In the HTRF ligand-binding assays, we label the receptors by covalently linking a Lumi4-Tb fluorophore to the N-terminal SNAP tag of the receptor. Since this fluorophore absorbs at 340 nm and emits (among others) at 620 nm, we can determine the relative receptor expression level in our cell samples. We have now included the expression levels relative to wild-type α_{1B} AR in Supplementary Table 5.

To obtain the amount of bound ligand, we determine the Förster resonance energy transfer between the Lumi4-Tb label linked to the receptor and an acceptor dye linked to the ligand, which is then given as the ratio between the emission of the acceptor and the donor fluorophore (Em520 nm / Em620 nm). This method not only gives us the desired HTRF ratio as a measure of bound ligand but also intrinsically normalizes the data to the receptor expression levels (i.e., Em620 nm).

We have now described the procedure to quantify and normalize expression levels in the Methods in lines 603–612.

3. Statistical analysis of the binding data is included in Table S6 at the end of the manuscript. It may be useful to mention this in the main figure legends in order to interpret the differences in the binding affinities of the ligands for different mutants.

We thank the reviewer for this suggestion. We have now mentioned the statistical analysis in Supplementary Table 6, and we referred to it in the legends of Figs. 4 and 5, and of Supplementary Fig. 3.

4. The observation on L334 is intriguing and worth looking into a little further. While there is a slight loss of thermostability on back mutation to F, how does it affect cyclazosin binding? What about the binding of other ligands?

To assess the effect of the F334L mutation on (+)-cyclazosin binding, we have carried out MD simulations on both the crystal structure (i.e., with L334) and a back-mutant with F334 (Fig. 2b and Supplementary Fig. 8). The simulations validated the crystallographically determined binding mode of (+)-cyclazosin, indicating that the F334L mutation does not significantly affect the ligand binding pose. (+)-cyclazosin displayed stable binding in both simulations.

Both F334 and L334 form comparable hydrophobic contacts with (+)-cyclazosin. We observed that F334 forms additional aromatic contacts with the quinazoline and furan rings of (+)-cyclazosin, as noted in lines 193–196 and Supplementary Table 3. We have shown that the difference in affinity between wild-type $\alpha_{1B}AR$ (i.e., with F334) and the crystallized construct (i.e., with L334 and additional engineering) is only marginal (see lines 117–119 and Supplementary Fig. 3d). The effect of the F334L mutation alone is thus expected to be even smaller.

For an early characterization, we carried out a radioligand binding assay and determined the affinity of 3H -prazosin — a ligand nearly identical to cyclazosin — for the stabilized mutant with L334 as well as for the L334F back-mutant. We observed a very similar affinity for the stabilized mutant with L334 compared to the L334F back-mutant, indicating only a minor impact of the mutation on inverse agonist binding.

Saturation-binding curves of 3H -prazosin for the stabilized $\alpha_{1B}AR$ mutant with L334 and for a back-mutant where L334 has been reverted to the wild-type phenylalanine residue. cpm, counts per minute. Binding experiments were performed on whole *E. coli* cells. Receptor expression levels were very similar.

Regarding the binding of other ligands, the F334L mutation possibly has a larger impact on the binding affinity of certain agonists. This is because F334 possibly forms an

aromatic “lid” on the agonist binding pocket, as mentioned in lines 416–417. However, such agonists are not the focus of the present study and we do not draw any new conclusions regarding agonist binding based on our crystal structure.

Reviewer #3 (Remarks to the Author):

Deluigi and coworkers present the first crystal structure of the alpha1B-adrenergic receptor (alpha1B-AR) in complex with an inverse agonist. Such structural information is essential to develop selective ligands targeting this GPCR, whose physiological and pathological roles are only partially characterized. The authors do a thorough comparison of the protein-ligand interactions observed for alpha1B-AR and other adrenergic receptor structures and propose residues 3.29 and 6.55 (Ballesteros-Weinstein numbering) as key determinants of ligand selectivity. Such prediction is further tested with site-directed mutagenesis data and ligand binding assays. The conclusions of the authors are well supported by the data and the study may be used as stepping stone to design selective ligands targeting alpha1B-AR. Therefore, I believe it merits publication, provided that a few minor issues are addressed.

We thank the reviewer for the appreciation of our study.

- On page 7, the authors characterize the alpha1B-AR structure as apparently captured in the inactive state. Besides the comparison of the TM6 position with other adrenergic receptor structures, I would suggest that the authors attempt to quantify the % degree active (as done in GPCRdb) or the activation index (for class A GPCRs, <https://ccda250.chemie.uni-erlangen.de/a100/a100-python/public/>).

We have now calculated the activation index (A^{100}) of the $\alpha_{1B}AR$ structure presented in this study and it corresponds to -45.7 , strongly indicating an inactive-state structure (Ibrahim, P., *et al.*, *J. Chem. Inf. Model.* **59**, 3938-3945 (2019)). We now state this in line 149.

To our knowledge, the GPCRdb structure browser does not yet offer the possibility to upload the coordinates of a to-be-released structure to determine the “% degree active”. The calculation will be performed by the GPCRdb once the coordinates have been released by the PDB, at the discretion of GPCRdb. Nonetheless, the distance between C α atoms of L87^{2.46} and L296^{6.37} in our structure corresponds to 11.0 Å, which translates to 0% active according to the cut-off chosen by the authors of the corresponding tool on GPCRdb (Kooistra, A., Christian Munk, C., *et al.*, PREPRINT (<https://doi.org/10.21203/rs.3.rs-354878/v1>)).

- On page 8, the authors describe a salt bridge potentially involved in activation. Do residues 7.36 and 2.65 correspond to one of the known GPCR activation microswitches?

(see e.g. <https://europepmc.org/article/ppr/ppr305890>) What are the equivalent residues in other adrenergic receptors (ARs) and, in case both active and inactive structures exist for those ARs, how do the interactions between these two residues change between the two states? Have mutations at these positions (either site-directed or natural variants) reported for ARs or other aminergic receptors, which also affect receptor activation?

We thank the reviewer for the suggestions. We have carefully reviewed the existing literature and came to the following conclusions:

(i) Residues 2.65 and 7.36 do not belong to the widely described microswitches of class A GPCRs, i.e., those involving the highly conserved residues of the P^{5.50}I^{3.40}F^{6.44} motif, the C^{6.47}W^{6.48}X^{P6.50} motif, the D(E)^{3.49}R^{3.50}Y^{3.51} motif, the N^{7.49}P^{7.50}XXY^{7.53} motif, as well as the water-mediated network involving D^{2.50} and other residues of the Na⁺ pocket.

(ii) There are four human GPCRs in addition to α_{1B} AR that bear the residue pair E^{2.65}-K^{7.36}, and they correspond to α_{1A} AR, α_{1D} AR, α_{2A} AR, and α_{2C} AR. In contrast, α_{2B} AR bears E^{2.65}-Q^{7.36}, and all three β ARs bear hydrophobic residues at both positions.

	Sequence-based ()	.65	.36
	Structure-based (GPCRdb)	x64	x35
[Human] α_{1A} -adrenoceptor		E	K
[Human] α_{1B} -adrenoceptor		E	K
[Human] α_{1D} -adrenoceptor		E	K
[Human] α_{2A} -adrenoceptor		E	K
[Human] α_{2B} -adrenoceptor		E	Q
[Human] α_{2C} -adrenoceptor		E	K
[Human] β_1 -adrenoceptor		V	V
[Human] β_2 -adrenoceptor		I	I
[Human] β_3 -adrenoceptor		A	L

(iii) To date, there are no other inactive- or active-state α_1 AR structures besides our inactive α_{1B} AR structure. In the inactive-state structure of human α_{2A} AR (PDB ID: 6KUX), the side chain of K^{7.36} has not been modeled. The same applies to the structure of human α_{2A} AR bound to the partial agonist RES (PDB ID: 6KUY). In the inactive-state structure of human α_{2C} AR (PDB ID: 6KUW), a salt bridge is observed between E^{2.65} and K^{7.36}, as in our structure. No active-state structures of α_{2A} AR and α_{2C} AR have been reported to date. In the active-state structure of human α_{2B} AR (PDB ID: 6K41), there is a hydrogen bond between E^{2.65} and Q409^{7.36}.

Overall, due to the lack of active-state structures for α_1 ARs, α_{2A} AR, and α_{2C} AR, and the sequence divergence of position 7.36 in α_{2B} AR, it is not possible to make a conclusive statement of how the interactions between residues 2.65 and 7.36 change upon activation. It seems, however, that an interaction is maintained in the active state, at least according to the available α_{2B} AR structure. Whether this applies to α_1 ARs as well remains to be determined.

(iv) Previous mutagenesis studies proposed that, in ligand-free (apo) α_{1B} AR, K331^{7.36} might form a salt bridge with D125^{3.32}, constraining the receptor in an inactive state. Upon agonist binding, the interaction formed between the positively charged nitrogen of the agonist and D125^{3.32} results in the disruption of the D125^{3.32}-K331^{7.36} constraint, initiating the activation process (Porter, J. E., *et al.*, *J. Biol. Chem.* **271**, 28318-28323 (1996); Porter, J. E., *et al.*, *Mol. Pharmacol.* **53**, 766-771 (1998); Porter, J. E. & Perez, D. M., *J. Biol. Chem.* **274**, 34535-34538 (1999); Porter, J. E. & Perez, D. M., *J. Pharmacol. Exp. Ther.* **292**, 440-448 (2000)).

Since our structure has a ligand bound, we cannot verify this hypothesis. We can only say that, upon inverse agonist binding, the side chain of K331^{7.36} forms a salt bridge with E^{2.65}.

Taken together, further mutagenesis, signaling, and structural studies are needed to verify the role of this salt bridge in the receptor activation process. Since this is beyond the scope of the present study, we believe that we should limit our statement to solely a description of what we see. We therefore made a change in lines 159–161, and we thank again the reviewer for drawing our attention to this point.

- On page 9, the authors explain that cyclazosin is likely to be protonated. If that is the case, I would suggest showing a protonated N1 in the 2D chemical representations of the ligand shown across the manuscript. Moreover, I understand from reference 49 that the calculated pKa of this molecule is 9. It might be worth mentioning this computational estimation to further support the consideration of a protonated molecule.

There is indeed strong agreement in the literature that N1 of the 2,4-diamino-6,7-dimethoxyquinazoline moiety is mostly protonated at physiological pH and that N1-protonation is relevant to receptor binding (e.g.: Campbell, S. F., *et al.*, *J. Med. Chem.* **30**, 49-57 (1987); Bordner, J., *et al.*, *J. Med. Chem.* **31**, 1036-1039 (1988); Matijssen, C., *et al.*, *J. Phys. Org. Chem.* **25**, 351-360 (2012)). We now show a protonated N1 in the 2D

chemical structures of cyclazosin and have added a sentence to the caption of Fig. 1. Nonetheless, we suspect that the computationally estimated pKa of 9 could be an overestimation, as the experimentally determined values for similar ligands were somewhat lower (pKa \approx 7–8). Therefore, we prefer not to explicitly mention the estimated pKa value of cyclazosin in our manuscript.

For consistency, we now show the analogous N1-protonation of prazosin and QAPB, both of which share the 2,4-diamino-6,7-dimethoxyquinazoline core with cyclazosin. The N1-protonation of prazosin and QAPB at physiological pH, as well as its relevance for receptor binding, are also well supported in the literature (e.g.: Chernyshev, V. V. *et al.*, *J. Pharm. Sci.* **93**, 3090-3095 (2004); Campbell, S. F., *et al.*, *J. Med. Chem.* **30**, 49-57 (1987); Bordner, J., *et al.*, *J. Med. Chem.* **31**, 1036-1039 (1988); Matijssen, C., *et al.*, *J. Phys. Org. Chem.* **25**, 351-360 (2012)).

Also, for consistency, we now show the protonated form of RS79948, whose N7-protonation is relevant for receptor binding (Qu, L. *et al.*, *Cell Rep.* **29**, 2929-2935 (2019); Chen, X. Y. *et al.*, *Cell Rep.* **29**, 2936-2943 (2019)). The same applies to the RS79948 analogs yohimbine and corynanthine (Supplementary Fig. 11). In all three ligands, the protonated nitrogen belongs to an aliphatic tertiary amine group with an estimated pKa > 9.

In Supplementary Figs. 1 and 3b, we nonetheless show the unprotonated forms of prazosin, cyclazosin, and QAPB, as the unprotonated 2D chemical structures of these ligands are still the most common representations encountered in literature. We have added a note in this regard to the figure captions.

- The authors describe how cyclazosin occupies the orthosteric binding site as well as secondary binding pockets. Do any of these secondary binding pockets correspond to the allosteric/vestibular binding site near ECL2 described for other class A GPCRs? In other words, can cyclazosin be considered a bitopic ligand that can "fill" simultaneously both the orthosteric and the allosteric binding sites?

To our knowledge, the terms "secondary", "extended", "minor", and "allosteric" have all been used to describe the ligand-binding site(s) near ECL2 observed in several aminergic GPCRs. Specifically, the term "allosteric binding site" has been used in this regard in several publications on muscarinic acetylcholine GPCRs to describe the pocket(s) targeted by some allosteric compounds as well as by bitopic ligands (e.g.: Keov, P. *et al.*,

J. Biol. Chem. **289**, 23817-23837 (2014); Thal, D. M. *et al.*, *Nature* **531**, 335-340 (2016); Gregory, K. J. *et al.*, *J. Biol. Chem.* **285**, 7459-7474 (2010)). The overlap between those “allosteric binding sites” and the “secondary binding pockets” described for other GPCRs has been confirmed before (e.g.: Michino, M. *et al.*, *Pharmacol. Rev.* **67**, 198-213 (2015)).

We preferred to stick to the term “secondary”, as defined by Michino *et al.*, *Pharmacol. Rev.* **67**, 198-213 (2015). Nonetheless, cyclazosin can indeed be considered a bitopic ligand. We have now added a corresponding statement in lines 179–181. In addition, we have also mentioned in lines 393–394 that the term “secondary binding pocket” sometimes corresponds to “allosteric binding site”.

- Do the authors observed any electron density that could correspond to one or more cholesterol molecules bound to alpha1-AR? (since cholesterol seems to be an allosteric modulator of other GPCRs, as nicely shown by some of the authors of this manuscript for the oxytocin receptor: <https://doi.org/10.1126/sciadv.abb5419>)

We thank the reviewer for this interesting suggestion. We have carefully re-inspected the maps. However, we do not observe any convincing electron density corresponding to cholesterol.

- The authors should provide more details regarding their docking protocol: was the receptor considered as rigid or flexible? (this could affect the clashes described in the manuscript); did they include information about the binding site in their docking protocol, in terms of initial position of the crystallographic ligand and/or putative binding residues?; how many docking poses were generated? was only the top scored pose analyzed or all of them?

We apologize to the reviewer for this oversight. Details of the docking experiments have now been added to the Methods in lines 651–660.

Regarding treating the receptor as rigid or flexible:

(i) In the initial docking experiments, $\alpha_2\text{cAR}$ was considered as rigid. These docking experiments suggests steric hindrance between L128^{3,29} and the methyl ester group of corynanthine. We recognize that this approach is limited by the rigidity of L128^{3,29} and the assumption that corynanthine adopts a very similar conformation and binding pose as yohimbine and RS79948. We have now clarified this point in the caption of Supplementary Fig. 11.

(ii) We thank the reviewer for suggesting flexible receptor docking as an alternate means of docking corynanthine into the $\alpha_2\text{C}\text{AR}$ structure. By allowing L128^{3.29} to be flexible, corynanthine was able to dock into a similar pocket as yohimbine, albeit with poorer docking scores. This allowed us to use MD simulations to probe the stability of yohimbine and corynanthine bound to $\alpha_2\text{C}\text{AR}$ and the role of L128^{3.29}. These data have now been included in Supplementary Fig. 11 and Supplementary Movie 1. The text in lines 309–314 has been modified accordingly. Importantly, the overall conclusions remain the same as in the initial submission.

In the MD simulations, the methyl ester group of corynanthine is sufficiently far away from L128^{3.29} to avoid a clash. However, due to the unfavorable stereochemical configuration of C5 compared to yohimbine, this comes at a cost: (i) ring flips are necessary to allow the methyl ester group to point away from L128^{3.29} and (ii) the ligand needs to adopt a “tilted pose” for most of the simulation time. A third factor is the backward movement of L128^{3.29} observed in the simulation with corynanthine but not with yohimbine. Nonetheless, despite sampling several conformations and positions within the pocket, corynanthine does not find a stable binding pose. In contrast, yohimbine displays very stable binding. The fact that L128^{3.29} is pushed back in the simulation with corynanthine suggests again that its bulky side chain is responsible for the unstable binding of corynanthine in $\alpha_2\text{C}\text{AR}$.

On a final note, MD simulations using an L128^{3.29}→A mutant resulted in unstable ligand binding. This is fully consistent with our findings revealing that L128^{3.29} is of major importance for high-affinity ligand binding to $\alpha_2\text{C}\text{AR}$.

- Regarding the molecular dynamics (MD) details, the authors ran simulations for both the crystallographic construct and a structure reverting the L334(7.39) to F mutation. Did they also revert the other thermostabilizing mutations introduced in the construct to facilitate crystallization (in particular the partial replacement of TM7 by the linker)?

Another mutation close to the ligand-binding pocket and adjacent to L334, namely, V333L, was also reverted in our MD simulations. No other mutations were reverted, as the primary goal of the simulations was to assess the effect of the F334L mutation on (+)-cyclazosin binding. The simulations validated the crystallographically determined binding mode of (+)-cyclazosin, indicating that the F334L mutation does not significantly affect the ligand binding pose.

No other mutations were directly involved in (+)-cyclazosin binding. Overall, the engineered construct used for crystallization had a very similar affinity for cyclazosin compared to wt $\alpha_{1B}AR$, indicating that the modifications did not substantially perturb cyclazosin binding (see lines 117–120). Specifically, the fusion of TM7 to the linker and crystallization chaperone had no significant effect on the affinity of cyclazosin. For more details, see also comments 2 and 3 of reviewer #1.

Regarding the analysis of the stabilizing mutations on overall stability by MD, we would consider this to be outside the scope of the present manuscript, and this may be the subject of a future study.

And how were the missing residues in the ICLs modeled?

MD simulations were performed in the absence of these residues. It is well known that these mostly unstructured regions are difficult to model, and since they are in all likelihood of insignificant consequence to the cyclazosin binding site, which was of primary interest in the present study, we chose not to build the missing residues in the ICLs into our models.

- I find commendable that the authors decided to run MD simulations to assess the impact of the F334(7.39) to L mutation. Considering their effort and the length of their simulations (~300 ns), I would suggest that they try to exploit this computational data for further analysis. Does the mutation affect the protein flexibility, in terms of RMSF? F334(7.39) is located one helical turn away from Y338(7.43), which in turns interact with the essential D125(3.32). Can the authors check whether there is some stacking interaction between F334 and Y338 and if the H-bond between Y338 and D125 (shown in Figure 2a) is maintained after reverting the mutation? I would also be curious to know more about the behavior of V197(45.52), whose side chain was only partially resolved in the X-ray structure.

We thank the reviewer for their commendation. The primary focus of our MD simulations was indeed to validate the binding pose of (+)-cyclazosin we observed.

- In our simulations there is very little difference between the backbone RMSF traces of the crystal structure and the back-mutant, so we think inclusion and discussion of these data is not warranted (see plot below).

- There is no direct π -stacking of F334 with Y338 due to the angle between these side chains. This can be now seen in Fig. 2b, where we have depicted the side chain of Y338 as well. Nonetheless, F334 and Y338 are sufficiently close in space to form other types of aromatic interactions (Martinez, C. R. & Iverson, B. L., *Chem. Sci.* **3**, 2191-2201 (2012)); however, back-mutation of L334 to F334 did not affect the conformation of the Y338 side chain or the hydrogen bond between Y338 and D125 (see below).

- The hydrogen bond between D125 and Y338 is maintained through the crystal structure simulation and is retained in the simulation of the L333V-L334F back-mutant. We have now added a statement to this regard in lines 196–199 and show the data in Supplementary Fig. 8.

- In the L333V-L334F model and MD simulation, the furan ring of (+)-cyclazosin makes an off-center aromatic interaction with the F334 aromatic ring, and it also occasionally (< 10% of simulation time) contacts the Y338 side chain through hydrophobic interactions. In the crystal structure and the corresponding simulation (with L334), the furan ring of (+)-cyclazosin preferentially also occupies this same stacked conformation with Y338 and is seen to rotate perpendicular to the aromatic amino acids occasionally. Therefore, the introduction of F334 does not substantially change the preference of the furan ring for

the stacked conformation. We have now added a statement to this regard in lines 196–199 and show the data in Supplementary Fig. 8.

- The reviewer is correct that the ECL2 residue V197 is only partially resolved in the electron density of our crystal structure. V197 has been previously implicated in mediating selectivity of ligands for α_1 ARs (Zhao, M. M., Hwa, J. & Perez, D. M., *Mol. Pharmacol.* **50**, 1118–1126 (1996)). In the MD simulations of the α_{1B} AR crystal structure, the side chain of V197 projects down into the top of the (+)-cyclazosin binding site but was free to rotate around χ_1 , potentially explaining the lack of electron density observed. The close proximity of V197 to the secondary binding pocket occupied by (+)-cyclazosin suggests it could play a role in ligand selectivity, but the structure and MD simulations do not allow further comment on this.

- I also liked very much that the authors honestly describe the possible limitations of the different modifications introduced in the crystallographic construct when describing the structure. The only thing not fully clear to me is how the linker inserted in place of the intracellular part of TM7 could affect the position of the adjacent TM6 (which in turn the authors used to describe the activation state of the receptor).

The position adopted by TM6 in our α_{1B} AR structure is very similar to the position of TM6 observed in other AR structures (Supplementary Fig. 6a, b — there is especially a high overlap with the carazolol- β_1 AR structure (PDB ID: 2YCW)). Moreover, the position of TM6 in our structure is similar to what observed in many other inactive-state class A GPCR structures as well. As a consequence, it appears unlikely that the linker or the crystallization chaperone induced an artificial conformation of TM6. Instead, TM6 probably adopted one of the naturally sampled conformations belonging to the inactive state.

The closed conformation of TM6 is consistent with the pharmacological properties of the ligand (an inverse agonist), stabilization of the receptor in an inactive state (see point 1 of reviewer #1), and the absence of a signaling protein. The classification of our structure

as an inactive-state conformation has now been confirmed by the activation index (see above). The activation index takes five distances into account, only one of which involves a residue in the intracellular portion of TM6 (Ibrahim, P., *et al.*, *J. Chem. Inf. Model.* **59**, 3938-3945 (2019)).

Furthermore, the linker replacing helix 8 and connecting the C-terminal end of TM7 to the crystallization chaperone turned out to lack regular secondary structure (Supplementary Fig. 2b), and thus, it probably displays a certain degree of flexibility. This flexibility probably reduces the likelihood that the linker forced TM6 to adopt an unnatural position.

As always in crystallography, it remains a possibility that protein engineering, the crystallization condition, or crystal packing favored the crystallization of a particular conformation over another. Nonetheless, in this case, we are confident that the crystallized conformation belongs to the ensemble of naturally sampled ones and agrees with the pharmacological properties of the ligand.

- Although probably beyond the scope of this manuscript, have the authors considered to use their MD trajectories to estimate the difference in ligand binding free energy between the XTAL and MD receptors? And use per-residue decomposition in order to e.g. see if residues 3.29 and 6.55 contribute not only to selectivity, but also significantly to binding free energy?

This is a very interesting suggestion but would clearly constitute an independent study all in itself. To derive free energies from MD trajectories, different set-ups must be used, and thus a whole new series of simulations. Furthermore, to arrive at acceptable error levels, the simulations would have to be even much more extensive. We thus believe that these calculations go far beyond the scope of this manuscript. Nonetheless, we thank the reviewer for the suggestion, which could well be the focus of a future study.

REVIEWER COMMENTS

Reviewer #1 (Remarks to the Author):

The authors addressed my initial comments and provided clarification and additional data that improves their manuscript. However, two key concerns still need to be resolved,

1. While somewhat improved, their data collection statistics are still borderline at best. I agree that "map quality" is more important than pure metrics, but perhaps the authors can make that point more clear to the non-GPCR-crystallographer, i.e. I) show additional panels with "nice" electron density in their figure S5, e.g. showing select side chain densities in more detail; II) comment explicitly in the main manuscript on the fact that their stats are lacking, and why they think their structural model is nonetheless valid; and III) discuss/contrast their "serial" approach with reference to an appropriate reference, e.g. the S1P5 structure of Hanson et al., Science (2012) and the microdiffraction data assembly method described therein; which seems to have integrated a comparable number of crystals into a higher quality dataset.

2. The authors show detailed functional evaluation of their receptor modifications in the rebuttal letter (their response to my Q1), but wish to not disclose it until an upcoming manuscript. Given the high number of modifications and use of non-mammalian expression system, careful structure validation is crucial. If they don't wish to provide these functional data, the authors should instead analyze how each individual point mutation influences the *inverse agonism* of their compound, and discuss their findings in the main text.

Reviewer #2 (Remarks to the Author):

The authors have satisfactorily addressed the comments and concerns raised on the original version of the manuscript. I recommend publication of the revised manuscript.

Reviewer #3 (Remarks to the Author):

The authors have addressed all the issues that I mentioned in my previous referee report, as well as most of the issues raised by the other reviewers. Therefore, I would like to recommend the paper for publication.

Only for the sake of curiosity, I would suggest that the authors further characterize the movement of the ligand in the binding site during the MD simulations. For the ligand RMSD plots shown in Figure S8b it is not clear which is the reference structure used to define RMSD = 0 angstroms and if the same reference structure was used to calculate the RMSD for both the crystallographic construct and back-mutant trajectories. Besides calculating the RMSD and the internal dihedral angles, the authors could add some metric that indicates the relative position of the ligand wrt the receptor, e.g. the distance of the center of mass of the ligand wrt to some invariant position of the receptor or the angle formed between the principal axis of the receptor and the long axis of the ligand.

REVISION ROUND 2

Responses to reviewers' comments are in blue.

Line numbering refers to the re-revised version of the manuscript.

REVIEWER COMMENTS

Reviewer #1 (Remarks to the Author):

The authors addressed my initial comments and provided clarification and additional data that improves their manuscript. However, two key concerns still need to be resolved,

1. While somewhat improved, their data collection statistics are still borderline at best. I agree that "map quality" is more important than pure metrics, but perhaps the authors can make that point more clear to the non-GPCR-crystallographer, i.e.

The poor data collection statistics are a consequence of the anisotropy correction of our data. Comparison with refinements using isotropic data extending to 3.1 Å resolution, however, clearly shows that the anisotropy-corrected data substantially improve the refinement and do not have any negative effects on the model.

We have now revised our manuscript, as explained below in detail.

I) show additional panels with "nice" electron density in their figure S5, e.g. showing select side chain densities in more detail;

In Supplementary Fig. 5, we show the electron density for the ligand and all the side chains relevant to the conclusions of the present study. We have now included the side chain of E106 in panel c as well as added ten new panels where the side chain densities are shown in more detail, as suggested by the reviewer. The criteria for showing these ten side chains in more detail are their particular importance for ligand binding and/or the fact that they are not perfectly visible in the overview pictures in panels c and d. The electron density unambiguously supports our structural model.

Here below, we have also reproduced Supplementary Fig. 5a–d using the 2Fo–Fc electron density map obtained from the 3.1-Å isotropically processed dataset. This map unambiguously agrees with the structural model based on the 2.87-Å anisotropy-corrected dataset, which is the structural model drawn here. Nonetheless, to reiterate, the

anisotropy-corrected data substantially improve the refinement and the overall quality of the maps.

The structural model based on the 2.87-Å anisotropy-corrected dataset unambiguously agrees with the 2Fo–Fc electron density map obtained from the 3.1-Å isotropically processed dataset (purple mesh). Cf. Supplementary Fig. 5a–d in the manuscript. The 2Fo–Fc electron density map is contoured at 1.0 σ . In the upper part of the figure, the two orientations modeled for the furan-2-yl-methanone substituent of (+)-cyclazosin are shown on the left and right, respectively. In the bottom part of the figure, residues belonging to the ligand-binding pocket and ECL2 are shown.

The focus of the present study is the ligand-binding site. Nonetheless, the quality of the electron density allowed modeling of most regions and side chains outside of the ligand-binding site as well. We now mention the receptor regions where the electron density is of lower quality in lines 138–140. These regions are either flexible or immediately preceding or following flexible regions, which likely accounts for the lower quality of the electron density.

II) comment explicitly in the main manuscript on the fact that their stats are lacking, and why they think their structural model is nonetheless valid; and

In response to the previous point, we have clarified that the electron density is actually of good quality, and it unambiguously supports our structural model.

Regarding the statistics, we understand that the reviewer wishes a more detailed explanation regarding their quality and suitability as a metric for our dataset.

We have now revised and added text to the Results section in lines 127–138. We now explain the structure determination process in more detail and clarify why our structural model is valid despite the low completeness and poor data collection statistics in the highest resolution shell. In addition, to reinforce this point and provide further clarification on the statistical indicators, we have now added text to the legend of Supplementary Table 3 and to the Methods section starting in line 578. The latter section now also includes a summary of the geometrical validation of our model.

We have also corrected the number of merged minisets, which is 27, and we apologize for this slight mistake.

III) discuss/contrast their "serial" approach with reference to an appropriate reference, e.g. the S1P5 structure of Hanson et al., Science (2012) and the microdiffraction data assembly method described therein; which seems to have integrated a comparable number of crystals into a higher quality dataset.

We thank the reviewer for this note. We have compared our approach with the suggested reference, and the two approaches are different in several aspects. Accordingly, we have now made some changes to our manuscript.

Compared to the data wedges collected by Hanson et al., comprising a maximum 6 degrees due to the rapid onset of radiation damage, our wedges are substantially larger and range between 15 and 32 degrees, thus rendering data reduction considerably easier. The range of the used data wedges is now specified in our text.

Furthermore, we removed the expression "serial crystallography" from our manuscript as this might cause confusion. In a "typical" serial crystallography approach, the quality of the data is strongly improved by merging many datasets from a (very) large number of crystals. In our case, the limited number of useful crystals required a careful inspection, evaluation, and processing of the available datasets, and only the best ones were merged. We have now specified this point and provided a more detailed description of our procedure in the manuscript (see above).

Finally, we selected the best partial datasets to be merged according to very similar cell parameters. While Hanson et al. used a different strategy, we believe that our approach

based on close morphology is more suitable in our case, as non-isomorphous crystals were possibly present in the sample (possibly as a consequence of slightly different conditions during crystallogenesis, slightly different handling of the samples, or simply because some of the partial datasets did not have a sufficient quality for unambiguous determination of the cell parameters). We have now specified this point in our manuscript.

2. The authors show detailed functional evaluation of their receptor modifications in the rebuttal letter (their response to my Q1), but wish to not disclose it until an upcoming manuscript. Given the high number of modifications and use of non-mammalian expression system, careful structure validation is crucial. If they don't wish to provide these functional data, the authors should instead analyze how each individual point mutation influences the *inverse agonism* of their compound, and discuss their findings in the main text.

We considered a functional characterization of each individual mutation to be outside the scope of our manuscript, as we describe an inactive-state structure. The reviewer has now alternatively suggested to analyze how each individual mutation influences the inverse agonism of the co-crystallized ligand. We thank the reviewer for suggesting this experiment; however, we believe that it is hardly feasible and would not lead to new insights, as explained below in more detail.

Consequently, we have now added the functional characterization of each individual mutation to the present manuscript, as initially wished by the reviewer (see Supplementary Fig. 3b, c, and Supplementary Table 2). We have now added the following text to the Results section in lines 112–115: “We observed that the following individual mutations substantially impair agonist-induced G_q signaling: S95^{2.54}→C, S150^{34.50}→Y, G183^{4.63}→V, D191^{ECL2}→Y, T295^{6.36}→M, V333^{7.38}→L, F334^{7.39}→L, and P349^{7.54}→L (Supplementary Fig. 3b, c, Supplementary Table 2).” Note that we had to rearrange Supplementary Fig. 3 and thus moved the chemical structure of QAPB to Supplementary Fig. 1.

Compared to the data previously shown in the rebuttal letter, we have now measured additional data points to increase data reliability.

We believe that the reader now becomes sufficiently informed about the effect of each individual mutation on receptor function. However, we do not wish to speculate in the

present manuscript on how some of the mutations impair signaling / stabilize the receptor in the inactive state. This would clearly constitute an independent study all in itself and would require a better knowledge of the receptor activation mechanism. Deciphering the receptor activation mechanism is clearly beyond the scope of this manuscript and will require further structural and computational studies, including the determination of an active-state $\alpha_{1B}AR$ structure bound to a G protein.

Regarding the alternative experiment suggested by the reviewer, we believe that it would not result in the desired insights for the following reasons: Inverse agonism is defined as the ability of a ligand to inhibit the receptor basal activity. The mutations that stabilize $\alpha_{1B}AR$ into the inactive state would in all likelihood abolish the receptor basal activity. Therefore, most likely, it would be impossible to measure the inverse agonism of cyclazosin, and thus the effect of the mutations on the inverse agonism itself. In addition, probably because wild-type (wt) $\alpha_{1B}AR$ has already a very low basal activity (Rossier, O., *et al.*, *Mol. Pharmacol.* **56**, 858-866 (1999); Hein, P., *et al.*, *Naunyn-Schmiedeberg's Arch. Pharmacol.* **363**, 34-39 (2001)), it is difficult to measure inverse agonism with the wt receptor. Instead, each mutation would have to be transferred into a constitutively active receptor variant and the experimental system would have to be established. However, to reiterate, the mutations that stabilize $\alpha_{1B}AR$ into the inactive state would in all likelihood abolish the receptor constitutive activity as well.

Reviewer #2 (Remarks to the Author):

The authors have satisfactorily addressed the comments and concerns raised on the original version of the manuscript. I recommend publication of the revised manuscript.

We thank the reviewer for this recommendation.

Reviewer #3 (Remarks to the Author):

The authors have addressed all the issues that I mentioned in my previous referee report, as well as most of the issues raised by the other reviewers. Therefore, I would like to recommend the paper for publication.

We thank the reviewer for this recommendation.

Only for the sake of curiosity, I would suggest that the authors further characterize the movement of the ligand in the binding site during the MD simulations. For the ligand RMSD plots shown in Figure S8b it is not clear which is the reference structure used to define RMSD = 0 angstroms and if the same reference structure was used to calculate the RMSD for both the crystallographic construct and back-mutant trajectories. Besides calculating the RMSD and the internal dihedral angles, the authors could add some metric that indicates the relative position of the ligand wrt the receptor, e.g. the distance of the center of mass of the ligand wrt to some invariant position of the receptor or the angle formed between the principal axis of the receptor and the long axis of the ligand.

For the RMSD values reported in Supplementary Fig. 8a, b, the reference structure at frame = 0 consists of either the crystallographic construct or the back-mutant after the MD relaxation protocol. While very similar, these structures are not exactly the same. We believe that this is the appropriate way to present such data to give an indication of the overall stability of each protein and of the ligand over time, which is what we are aiming to give. We have now better clarified this point in the Methods section in lines 705–706.

To better indicate the relative position of the ligand with regard to the receptor, we have now added panels h–m to Supplementary Fig. 8. These panels show additional receptor-ligand distances and their frequency distributions throughout both simulations. In

addition, we have now included in panels c and d the distance between L334 and the quinazoline and furan rings of (+)-cyclazosin, respectively. These distances, together with the dihedral angles, further confirm that the ligand adopts a very similar and stable binding pose both in the presence and absence of the back-mutations (as also explained in response to point 6 of reviewer #1 in the first revision round).

REVIEWERS' COMMENTS

Reviewer #1 (Remarks to the Author):

The authors have now satisfactorily addressed my concerns, leading to a much improved manuscript that I can recommend for publication.

Reviewer #3 (Remarks to the Author):

The authors have addressed the minor comments that I included in my previous referee report, regarding the further analysis of the molecular dynamics simulations. Therefore, I would like to recommend the paper for publication.

REVIEWERS' COMMENTS

Reviewer #1 (Remarks to the Author):

The authors have now satisfactorily addressed my concerns, leading to a much improved manuscript that I can recommend for publication.

We thank the reviewer for the constructive inputs and this recommendation.

Reviewer #3 (Remarks to the Author):

The authors have addressed the minor comments that I included in my previous referee report, regarding the further analysis of the molecular dynamics simulations. Therefore, I would like to recommend the paper for publication.

We thank the reviewer for the constructive inputs and this recommendation.